# Single-species dinoflagellate cyst carbon isotope fractionation in core-top sediments: environmental controls, CO₂-dependency and proxy potential

Joost Frieling[1‡], Linda van Roij[1], Iris Kleij[1], Gert-Jan Reichart[1,2] & Appy Sluijs[1]

1. Department of Earth Sciences, Faculty of Geosciences, Utrecht University, Princetonlaan 8, 3584CB Utrecht, The Netherlands

2. NIOZ Royal Netherlands Institute for Sea Research and Utrecht University, Texel, The Netherlands

‡ now at: Department of Earth Sciences, University of Oxford, South Parks Road, Oxford, OX1 3AN, Oxford, United Kingdom

*Correspondence to*: J. Frieling (joost.frieling@earth.ox.ac.uk)

**Abstract.** Sedimentary bulk organic matter and various molecular organic components exhibit strong $CO_2$-dependent carbon isotope fractionation relative to dissolved inorganic carbon sources. This fractionation ($\varepsilon_p$) has been employed as proxy for paleo-$pCO_2$. Yet, culture experiments indicate that $CO_2$-dependent $\varepsilon_p$ is highly specific at genus and even species level, potentially hampering the use of bulk organic matter and non-species-specific organic compounds. In recent years, significant

progress has been made towards a $CO_2$ proxy using controlled growth experiments with dinoflagellate species, also showing highly species-specific $\varepsilon_p$ values. These values were, however, based on motile specimens and it remains unknown whether these relations also hold for the organic-walled resting cysts (dinocysts) produced by these dinoflagellate species in their natural environment. We here analyze dinocysts isolated from core-tops from the Atlantic Ocean and Mediterranean Sea, representing several species (*Spiniferites elongatus, S. (cf.) ramosus, S. mirabilis*, *Operculodinium centrocarpum* sensu Wall

& Dale (1966) (hereafter referred to as *O. centrocarpum*) and *Impagidinium aculeatum*) using Laser ablation – nano Combustion – Gas Chromatography – Isotope Ratio Mass Spectrometry (LA/nC/GC-IRMS). We find that the dinocysts produced in the natural environment are all appreciably more $^{13}C$-depleted compared to the cultured motile dinoflagellate cells, implying higher overall $\varepsilon_p$ values and, moreover, exhibit large isotope variability. Where several species could be analysed from a single location, we often record significant differences in isotopic variance and offsets in mean $\delta^{13}C$ values between

species, highlighting the importance of single-species carbon isotope analyses. The most geographically expanded dataset, based on *O. centrocarpum*, shows that $\varepsilon_p$ correlates significantly with various environmental parameters. Importantly, *O. centrocarpum* shows a $CO_2$-dependent $\varepsilon_p$ above ~240 μatm $pCO_2$. Similar to other marine autotrophs, relative insensitivity at low $pCO_2$ is in line with active carbon concentrating mechanisms at low $pCO_2$, although we here cannot fully exclude that we partly underestimated $\varepsilon_p$ sensitivity at low $pCO_2$ values due to the relatively sparse sampling in that range. Finally, we use the

relation between $\varepsilon_p$ and $pCO_2$ in *O. centrocarpum* to propose a first $pCO_2$ proxy based on a single dinocyst species.

## 1 Introduction

Stable carbon isotope fractionation in marine autotrophs is governed for a large part by the carbon fixing enzyme RubisCO (e.g. Farquhar et al., 1989; Roeske and O'Leary, 1984), which implies most marine organic matter and therefore sedimentary marine organic matter is strongly $^{13}$C-depleted with respect to the dissolved inorganic carbon (DIC) source ($CO_2$ (aq), $HCO_3^-$ or $CO_3^{2-}$), with the stable carbon isotope composition ($\delta^{13}C$) of organic matter ranging from -10 to -30‰ (Freeman and Hayes, 1992). While many groups of marine autotrophs show clear $CO_2$-dependent carbon isotope fractionation ($\varepsilon_p$), the exact relation strongly varies between marine phytoplankton groups, genera and cell morphologies (Popp et al., 1998; Boller et al., 2011, 2015; Brandenburg et al., 2022). Still, because of the assumed $CO_2$-dependency of RubisCO fractionation, bulk marine organic matter and more specific organic compounds of marine autotrophs (e.g. lipids biomarkers) have been proposed and applied as $pCO_2$ proxies over the past decades (Freeman and Hayes, 1992; Naafs et al., 2016). The application of these $pCO_2$ proxies (e.g. Bijl et al., 2010; Pagani et al., 2011; Schoon et al., 2011; Witkowski et al., 2018) has provided constraints on past atmospheric $pCO_2$ and earth system sensitivity beyond the ice core record (e.g. Pagani et al., 2010; PALAEOSENS, 2012). However, many of the organic compounds used for $CO_2$ reconstructions such as alkenones (e.g. Pagani, 2013), phytane (e.g. Witkowski et al., 2018), porphyrins (e.g. Freeman and Hayes, 1992) or bulk organic matter (e.g. Hayes et al., 1999) are not unique to a single species, genus and sometimes not even a group of organisms. This implies that reconstructions based on these compounds integrates interspecific differences in $CO_2$-dependency, which complicates the interpretation of such proxy records. Secondly, even if specific compounds derive from a single species or genus, they intrinsically derive from a multitude of individual organisms, differing in shape and size, affecting isotopic fractionation and hence limiting the accuracy of such $CO_2$ reconstructions.

Part of the uncertainties and biases in carbon isotope fractionation can be circumvented if the carbon isotopic fractionation of individual fossils can be analyzed. In recent years, significant progress has been made towards a $CO_2$ proxy based on the stable carbon isotope fractionation in organic walled dinoflagellate cysts (Burkhardt et al., 1999; Hoins et al., 2015, 2016a, 2016b; Wilkes et al., 2017). A fraction (~15%) of modern dinoflagellates produces an organic resting cyst or dinocyst as an obligatory part of their lifecycle (Evitt, 1985). The organic resting cysts from autotrophic species have excellent preservation potential, are often highly oxidation-resistant (Zonneveld et al., 1997, 2019; Kodrans-Nsiah et al., 2008) and several ubiquitous extant genera and species, such as *Spiniferites* spp. and *Operculodinium centrocarpum*, have extremely long geological records (Fensome et al., 1996; Williams et al., 2004). The ecology and morphology of these long-ranging species seemingly remained unchanged for millions of years (Frieling and Sluijs, 2018). Most importantly, recent advances in methodology allow for analyses of species-specific single-cyst $\delta^{13}C$ (van Roij et al., 2017; Sluijs et al., 2018). This presents the opportunity to quantify environmental controls on $\varepsilon_p$ of individual dinoflagellate cysts and hence species to assess the potential to obtain more accurate paleo-$pCO_2$ estimates from sedimentary records.

Controlled growth experiments across a range of $CO_2$ levels representative for the last glacial (e.g. Barnola et al., 1987), modern and future carbon emission scenarios (Eberlein et al., 2016; Hoins et al., 2016a, 2016b, 2015; IPCC, 2014; Rost et al.,

2006; Van de Waal et al., 2013; Wilkes et al., 2017) showed species-specific $CO_2$-dependent $\varepsilon_p$ for multiple dinoflagellate
species. From these, the species *Protoceratium reticulatum* and *Gonyaulax spinifera* (Hoins et al., 2015, 2016a, 2016b) are of
particular interest as these produce the organic cyst species *Operculodinium centrocarpum* sensu Wall and Dale, (1966),
hereafter referred to as *O. centrocarpum,* and *Spiniferites (cf.) ramosus,* hereafter referred to as *S. ramosus*, respectively (Head,
1996; Zonneveld et al., 2013). These cyst species have their first occurrences in the geological records around ~60 and 130
million years ago (Ma), for *O. centrocarpum* and *S. ramosus*, respectively (Williams et al., 2004), thus providing potential for
deep-time $p$CO$_2$ reconstructions.

Before $\varepsilon_p$ values based on dinocysts can be used for reconstructing $p$CO$_2$, several fundamental questions need to be addressed.
Although $\delta^{13}C_{DIC}$ exerts a major control on dinocyst $\delta^{13}C$ (Sluijs et al., 2018), it remains uncertain whether the $CO_2$ control on
$\varepsilon_p$ of motile cells from controlled growth experiments can be translated to their cysts formed in the natural environment. In
addition, potential offsets in $\varepsilon_p$ values between the motile cells and the cysts need to be established. This is especially important
as the cell-cyst relations in carbon isotope ratios are not necessarily straightforward, because bulk biomass such as cysts
potentially deviates in $\delta^{13}C$ values from the various cell components and potentially not by a constant offset (e.g. Freeman,
2001; Hayes, 2001; Pancost and Pagani, 2006; Schouten et al., 1998; Van de Waal et al., 2013; Wilkes et al., 2018).

We here present the first core-top data for single-species dinocysts to constrain the environmental controls on $\varepsilon_p$. We focus on
the species *O. centrocarpum* and compare this data, when possible, to several species of *Spiniferites* (*S. ramosus, S. mirabilis,
S. elongatus*) and *Impagidinium aculeatum*. The established environmental relations are subsequently evaluated using simple
models converting carbon isotope fractionation in dinocysts into $p$CO$_2$ values for the surface waters.

### 2.1 Materials

The primary dataset is based on 34 core-top samples (Table 1, Fig. 1), collected from the North Atlantic Ocean and
Mediterranean Sea, with a secondary dataset based on pre-industrial sample material spanning ~0 – 1500 common era (CE)
from ENAM9606 the North Atlantic (Richter et al., 2009). The core-top samples encompass a substantial natural $p$CO$_2$ (aq)
gradient because the rate of cooling of the North Atlantic Current exceeds that of $CO_2$ uptake, whereas $p$CO$_2$ in the
Mediterranean is close to or slightly above equilibrium with the atmosphere. Sample selection is further based on the dinocyst
occurrence maps of Zonneveld et al. (2013), including only samples with an expected relative abundance of at least 10-20%
of the target species. Similarly, the coverage of environmental parameters such as sea surface temperature (SST) and $p$CO$_2$
and difference in environmental settings was maximized during sample selection. Existing ocean databases are used for
obtaining the relevant environmental parameters (Table 1).

### 2.2 Methods

Using standard palynological techniques (see e.g. Brinkhuis et al., 2003), ca. 5–10 g freeze-dried sediment of the
upper 1–2 cm of core material was processed for each sample. This involved dissolving carbonates and silicate components

using strong acids (HCl, 30% and HF, 38–40%). After acid steps, residues were pH-neutralized and sieved using an ultrasonic bath and 250 and 10μm nylon mesh sieve to remove large and small particles, respectively. Subsequently, samples were transferred to glass test tubes with demineralized water and centrifuged at 3200 rpm for 10 minutes to obtain an optimum concentrate of the sample material. Prior to dinocyst selection, samples were stored in 4 mL glass vials in demineralized water.

A micro-manipulator consisting of a Leica inverted microscope and a Narishige IM-9B microinjector connected to a strung-out pipette was used to manually select individual dinocysts from a water droplet on a glass petri dish. Dinocyst selection followed a strict protocol, in which cyst morphology (primarily cyst shape and process length) was kept constant and contribution of other organic particles minimized. Specimens with darker coloration or amorphous organic matter adhered to the cyst or processes were avoided. In the case of *O. centrocarpum*, the morphological selection primarily involved selecting specimens of equal size and process length to avoid cysts that may be derived from different environments (e.g. Mertens et al., 2009). For *Spiniferites*, we were able to distinguish and separate three distinct morphological species in sufficient numbers; *S. ramosus*, *S. elongatus* and *S. mirabilis*. For all dinocyst species, the selected diameter excluding processes was in the order of ~30–40 μm, except for *S. mirabilis* (~60 μm), although constraining the exact size of each individual specimen was not feasible within the current analytical procedures. Stable carbon isotope analyses for individual samples are based on replicating the analyses of single of dinocysts, with ~30 individual measurements being conducted to obtain a reasonably precise (~0.3–0.4‰) sample average (van Roij et al., 2017). Given the size of the dinocysts used here (~30–40 μm cyst diameter), 3–7 specimens were required for each measurement and hence ~150 cysts were required to obtain sample averages (Table 1).

Dinocysts were placed on a 6 mm Ø nickel sample tray, after which an identical second tray is added on top and compressed to fixate the dinocysts. Before placement in the ablation chamber, approximately ~1 $mm^2$ of International Atomic Energy Agency CH-7 (IAEA-CH7) polyethylene standard (PE; certified $\delta^{13}C$ value -32.15‰ ± 0.05‰; 1σ) was added to the sample tray. Stable carbon isotope analyses of the dinocysts followed the procedures described in previous work (van Roij et al., 2017; Sluijs et al., 2018), utilizing the recently developed Laser Ablation – nano Combustion – Gas Chromatography – Isotope Ratio Mass Spectrometry (LA-nC-GC/IRMS) method. Fragments resulting from deep ultraviolet LA were carried using a continuous Helium flow in 0.32 mm capillaries and oxidized in a combustion oven at 940 ºC. The resultant $CO_2$ was transported to a GC combustion interface, dried in a nafion tube using a He counterflow and subsequently into a ThermoFisher DeltaV Advantage IRMS for isotope analysis. Each analytical run included 5 standards with signal intensity above 4 Vs (ca. 40 ng C; $\delta^{13}C$ precision better than 0.5‰) to allow calibrating to the Vienna Peedee Belemnite (VPDB) scale. Direct visual monitoring of the ablation process was used as initial quality assessment of each individual measurement.

To calculate the fractionation factor $\varepsilon_p$ of the dinocysts relative to dissolved inorganic carbon (DIC) from which the dinocyst was produced, we take the $\delta^{13}C_{DIC}$ from the modeled grid of Tagliabue and Bopp (2008). As many dinoflagellate species, including those that produce *O. centrocarpum* and *S. ramosus* cysts, are able to utilize both $HCO_3^-$, which makes up the majority of DIC, and $CO_2$ for carbon fixation (Hoins et al., 2016a), we also compare the $\delta^{13}C_{DINO}$ data to $\delta^{13}C_{CO2}$ and with overall sea water carbon partitioning.

$\varepsilon_{p\text{-}DIC}$ is calculated as: $\delta^{13}C_{DIC} - \delta^{13}C_{DINO}$ and $\varepsilon_{p\text{-}CO2}$ is calculated as $\delta^{13}C_{CO2} - \delta^{13}C_{DINO}$. For the latter the $\delta^{13}C$ of dissolved $CO_2$ is calculated from $\delta^{13}C_{DIC}$ using the temperature-dependent fractionation between DIC and $CO_{2(aq)}$ (Mook et al., 1974). To evaluate the dominant contributions to $^{13}C$-fractionation in dinocysts, we compare the $\varepsilon_{p\text{-}DIC}$ and $\varepsilon_{p\text{-}CO2}$ values to measured and interpolated physicochemical parameters. We test a suite of parameters, $[NO_3^-]$, $[PO_4^{3-}]$, $[Si]$, alkalinity, $pCO_2$, SST and SSS, which are extracted using Ocean Data View (https://odv.awi.de/) from existing (gridded) datasets (Gouretski

and Koltermann, 2004; Takahashi et al., 2014, 2016) (Supplementary Data File). Where possible, data are averaged over a grid 4º longitude and latitude around the sample position. This is both to reduce errors introduced by data scarcity and to account for potential lateral transport of dinocysts during sinking (Nooteboom et al., 2019). Carbonate chemistry is calculated using the R-package `seacarb` (Gattuso et al., 2019), with alkalinity and $pCO_2$ as input variables to calculate the other relevant parameters of the carbonate system: the relative contributions of $CO_2(aq)$, $HCO_3^-$ and $CO_3^{2-}$, i.e. carbon speciation.

Ideally, all environmental parameters would be known for the different locations, as well as the time the dinoflagellates lived and encysted. This is, however, unfeasible because the dinocysts assemblage in core-top sediments (typically the upper 2 cm of sediment) integrates conditions over several centuries, assuming moderate to low average sediment accumulation rates ($< 10$ cm kyr$^{-1}$) that generally characterise open ocean settings such as examined here. We therefore apply a rough correction for $pCO_2$, based on the assumption that local air-sea gas exchange has remained similar, that equals the

atmospheric $pCO_2$ rise between the sampling date and 1850 CE. The correction from actual measurements to 'pre-industrial' conditions for $pCO_2$ yields a substantial offset due to the ~90 ppmv atmospheric $pCO_2$ rise from 1850 CE to the average sampling date (*ca.* 2000 CE). As this correction is broadly similar for all sample localities, it has only a small impact on the overall pattern in the $CO_2$ data (Fig. 3). We employed a Monte Carlo simulation to assess the potential impact of the $pCO_2$ correction by propagating (1) the 5% analytical error on $pCO_2$ values and add $+45 \pm 15$ ppm to reflect a normally distributed

mixture of modern and pre-industrial conditions and (2) a resampled uncertainty derived from the $pCO_2$ rise since pre-industrial times (1800 – 2000 CE). Both scenarios are set-up to represent worst-case scenarios; a single error drawn from the error distribution is imposed on a sample basis and not a resampled average of the number of $\delta^{13}C_{DINO}$ measurements within a sample, as that would reduce the error through averaging.

        Changes in SST, SSS and nutrient concentrations are also expected, partly also by anthropogenic activity, but offsets

in these parameters are generally subtle and more local compared to the changes in $pCO_2$ and hence would require site-specific reconstructions. Still, recent wide-spread eutrophication and enhanced productivity may impact the carbon isotope results through increased DIC uptake in algal blooms (i.e. counteracting the impact of enhanced $pCO_2$). However, as eutrophication mainly affected coastal areas (Hallegraeff, 1993; Anderson et al., 2002), this is expected to play a minor factor at our, mostly open marine, sample localities (Fig. 1). Lastly, long-term natural changes in nutrients, SSS and SST also occur, and it is

currently not possible to fully filter out the various anthropogenic offsets. With the exception of $pCO_2$, we hence assume all parameters (SST, SSS, nutrients) to have remained constant over the period the core top samples represent.

## 3. Results

### 3.1 Carbon yields from dinocyst analyses

Despite our pre-screening to include only samples with high relative abundances of the target species, some of the selected samples contained too few dinocysts or in too low abundance relative to other organic particles (amorphous organic matter, plant debris, pollen, non-dinocyst marine palynomorphs etc.), to be used in our study. Ultimately, out of the initial core-top sample set of 34 samples, 19 were found suitable for species specific dinocyst stable carbon isotope analyses (Table 1). Typically, ~150 individual cysts were picked and analyzed for a total of 20–50 measurements, amounting to 3–7 cysts per

carbon isotope measurement. We calculate an average signal size of 0.2 Vs for a single cyst, which amounts to a carbon yield of ~6–7 ng C cyst$^{-1}$ (van Roij et al., 2017). Although the variability in signal intensity from individual measurements suggests there is substantial intra-sample (cyst-cyst) variability, no significant offsets in average carbon content per cyst were observed between samples, suggesting the average carbon content of the cysts within each of the analyzed populations is similar. *Spiniferites mirabilis* is the notable exception to this rule, as far fewer specimens of this species are needed for a single $\delta^{13}C_{cyst}$

measurement. Based on the signal intensity per specimen we estimate that this larger cyst species contains twice the amount of C compared to *S. ramosus, S. elongatus* and *O. centrocarpum*.

### 3.2 Carbon isotope data

3.2.1 Signal Intensity

The 949 individual core-top analyses range in $\delta^{13}C$ from ~ -18.5 ‰ to -35.5 ‰ (all $\delta^{13}C$ are relative to Vienna Peedee Belemnite (VPDB)), and the 137 down-core analyses range from ~ -19.4 ‰ to -28.1 ‰. No relation was observed between $\delta^{13}C$ and signal size (Vs), except at the very low end ($\leq$0.2 Vs) (Fig. 2), in line with earlier analyses (van Roij et al., 2017). In this low range, the median of the $\delta^{13}C$ values rises from -28 ‰ below 0.1 Vs to values between -22 and -25 ‰ above 0.2 Vs. In the $\leq$0.2 Vs range the $\delta^{13}C$ average of both the cysts and PE converge between -25 ‰ and -30 ‰, with substantial scatter. Poorer

performance at such low C masses and signal intensities is expected, as these extremely small signal sizes and poor signal to noise ratio (below ~3:1) approach the limit of our method. Consequently, even a very minor contamination source would bias values and result in larger scatter, as also apparent in the PE standard at a similar signal intensity (van Roij et al., 2017; Fig. 2). Due to a worsening signal-noise ratio, we find a noticeable degree of $\delta^{13}C$ biasing from a background C source within the system is likely to occur at signal intensities $\leq$0.5 Vs and are particularly pronounced $\leq$ 0.2 Vs, and values for the standard and

dinocysts converge around -27 ‰ in this range (Fig. 2A). A similar background $\delta^{13}C$ value was also obtained after liquid nitrogen trapping (van Roij et al., 2017). The source of this C remains elusive. It is unlikely to be related to the ablation (etching) of the nickel plate or associated with the water used to pick sample from, as measured blanks for those result in much lower signal intensities and neither source would affect the measurements of the PE standard. Lastly, a significant contribution of atmospheric $CO_2$ ($\delta^{13}C$ around -8‰) appears unlikely due to the $\delta^{13}C$ signature of the background signal (-27 ‰). Though

the origin of the background C remains unknown, we can use the trapping experiment to estimate the relevant background contribution (van Roij et al., 2017). We calculate the typical contribution is likely between 0.024 and 0.08 Vs, given a

background C flux of 0.0008 Vs per second (van Roij et al., 2017) and typical duration of measurements (30–50 s for $\delta^{13}C_{DINO}$ and up to ~100 s for PE standard).

Before comparing our data with environmental variables, we therefore assess the impact of a very minor, but consistent, background contamination on the carbon isotope signal at low signal intensities (e.g. Fig. 2A). We find that a constant addition of *ca.* 0.04 Vs ($\leq$ 1 ng C) of a background C source with a $\delta^{13}C$ of -27 ‰ can explain the positive skewing in the standard PE $\delta^{13}C$. Using a simple isotope endmember / mass balance mixing model to correct for skewing (Fig. 2B), we calculated an average deviation from the measured PE and dinocyst values for intensities below 0.2 Vs in the order of |2.6 ‰| and |1.3 ‰|, respectively. The standard deviation of the data increases approximately 3-fold (Fig. 2B) compared to the raw measurement data below 0.2 Vs, but remains virtually unchanged above 0.2 Vs and the calculated deviation from the measured value is also much reduced above 0.2 Vs (|<0.3 ‰|).

The data correction using our simple mixing model eliminates the skew towards -27 ‰ at low signal intensities, and removes signal size $\delta^{13}C$-dependency below 0.2 Vs for both the isotopically homogeneous PE and the heterogeneous dinocyst data (Fig. 2A, B). This suggests our method of bias correction is warranted, but the increased variability at very low intensities and lack of independent control on the exact size and $\delta^{13}C$ of the background contamination implies the data associated with the lowest signal intensities remain significantly less reliable. We therefore apply a conservative cut-off, and use only corrected data with a signal size above 0.2 Vs.

### 3.2.2 Skewed distributions and outlier omission

The drift-corrected $\delta^{13}C_{DINO}$ is non-normally distributed in many core-top samples and also in different species (Table 1, Fig. 4). Distributions tend to be tailed towards lower values, exaggerated by the presence of a small amount of outlier values. This is not due to analytical error or otherwise directly related to low signal intensity as we used a 0.2 Vs cut-off to eliminate samples with potentially unreliable signal-noise ratios (see above) and a minor correction for background C addition was sufficient to eliminate skewing at low signal intensities. The absence of such signals in the down-core samples (Supplementary Fig. 1) suggests that the outliers and skewing in the sampled core-top populations could represent a real signal.

Based on typical deep ocean sedimentation rates in the range of centimetres per kyr, the core-top samples are expected to contain a mixed assemblage of dinocysts produced mostly within the last centuries to millennia but could also include cysts produced during the last few decades that are likely affected by anthropogenic influences. It is particularly relevant to consider because a steep $\delta^{13}C$ decrease (~2‰ since 1850 CE of which >1.5‰ occurs after 1950 CE) (Francey et al., 1999; Keeling et al., 2017) accompanies the $p$CO$_2$ rise (>130 ppmv since 1850 CE, of which >100 ppmv after 1950 CE). So even if enhanced carbon isotope fractionation at higher $p$CO$_2$ (Freeman and Hayes, 1992; Hoins et al., 2015; Brandenburg et al., 2022) would not play a role, the most recent specimens are likely to be impacted by decreasing $\delta^{13}C_{DIC}$.

As age integration in the modern era may result in a mixture of cysts representing a range of environmental conditions, especially with respect to CO$_2$ concentrations and $\delta^{13}C_{DIC}$, it is important to consider the potential age-distribution of dinocysts

before comparing $\delta^{13}C_{DINO}$ and $\varepsilon_p$ to environmental variables. In an ideal scenario, cysts produced after 1850 CE should be avoided in proxy-calibration efforts to eliminate a systematic bias towards the most recent times when atmospheric $CO_2$ was already elevated above pre-industrial Holocene background (~280 ppmv). Because an accurate age-correction for the Suess-effect is technically unfeasible because the age-distribution of $\delta^{13}C_{DINO}$ measurements cannot be constrained, we illustrate the influence of $\delta^{13}C_{DINO}$ data treatment (Fig. 3) and use Monte Carlo simulations of different error distributions to test and the potential impact of the $pCO_2$ correction. We also compared both measured $pCO_2$ and $pCO_2$ around 1850 CE (see section 2.2) to $\varepsilon_p$ calculated using both our raw $\delta^{13}C_{DINO}$ data and the $\delta^{13}C_{DINO}$ data after drift-correction and removal of statistical outliers identified within the sample-specific single species populations. This final step of data-treatment removed positive and negative measurement outliers from the sample- and species-specific $\delta^{13}C$ population (outside $\pm 2.5$ IQR), after eliminating the extremely low-signal intensities (<0.2 Vs) and correcting for the drift induced by background C in the system.

Altogether, out of a 949 core-top measurements, we omit 43 measurements with signals < 0.2 Vs and 24 statistical outliers, which leaves 882 individual $\delta^{13}C_{DINO}$ measurements, 560 for *O. centrocarpum*, 293 for *Spiniferites* (158 *S. ramosus*, 69 *S. elongatus* and 66 *S. mirabilis*) and 29 for *I. aculeatum* (Table 1). Most of the 67 omitted measurements have comparatively low $\delta^{13}C$ and the resulting $\delta^{13}C$ of the populations are close to statistically indistinguishable from a normal distribution (Shapiro-Wilk $p = 0.05–0.1$) or representative of a normal distribution (Table 1). Although the data-treatment partly removed the negative skew on the $\delta^{13}C_{DINO}$ distribution (Table 1), the combined effects of drift-correction and outlier-removal on sample / species-mean $\delta^{13}C_{DINO}$ are generally small (Fig. 3). This is expected as drift-correction averages only ~0.25 ‰ and the negative and positive outliers represent only a small percentage of the total measurements (Table 1).

Distinctly non-normally distributed $\delta^{13}C$ values were not previously observed in recent pollen and ancient dinocyst species analyzed with the same method (van Roij et al., 2016; Sluijs et al., 2018). The here presented down-core pre-industrial $\delta^{13}C_{DINO}$ show a similar mean, variance and data distribution to the nearby core-top samples (Supplementary Fig. 1), suggesting that, at least for these nearby localities, the analysed core-top specimens represent pre-industrial conditions. We find an influence of Suess-effect and increased $pCO_2$ impacts on the $\delta^{13}C_{DINO}$ data is the most likely factor to explain the appearance of a small number of predominantly [13]C-depleted outliers and resulting (subtle) negative skewing of the $\delta^{13}C$ distributions (Fig. 4).

We use the background and outlier-corrected $\delta^{13}C_{DINO}$ data and compare these data with $CO_2$ conditions prevalent around 1850 CE (Fig. 6) and explore the effects of age-integration by propagating different error distributions representative of the $pCO_2$ change since 1850 CE (Fig. 7B – D). For practical purposes, we assume all $\delta^{13}C_{DINO}$ populations to be normally distributed for further statistical analyses. We then use the mean carbon isotope value ($\delta^{13}C_{DINO}$) and signal intensity in volt seconds (Vs) of each sample. The standard error of the mean ranges from ~0.2 to 0.7‰ and depends primarily on the number of measurements in cases where n<30, in line with expected values based on replicate measurements of the PE standard (van Roij et al. 2017) (Fig. 5).

### 3.3 Environmental parameters and correlation

The range of measured $\delta^{13}C_{DINO}$ values (~5‰) far exceeds the variability in surface ocean $\delta^{13}C_{DIC}$ (~1 ‰) and $\delta^{13}C_{CO2}$ (~2.5‰), implying the observed range likely reflects differences in fractionation related to changing uptake or leakage of different inorganic carbon phases ($CO_2$ and $HCO_3^-$; Sharkey and Berry, 1985; Hoins et al., 2016a), and this hence determines most of the variability in the $\delta^{13}C_{DINO}$ data. Here, we quantitatively assess fractionation as a function of several environmental parameters.

The simple (non-weighted) linear regressions show poor correlations between $\varepsilon_{p\text{-}DIC}$ and $\delta^{13}C_{DIC}$ for all environmental parameters, and the correlations slightly improve when compared to $\varepsilon_{p\text{-}CO2}$ (Table 2, 3). However, none of the tested parameters individually explain the majority of the observed variance in $\varepsilon_{p\text{-}DIC}$ (maximum $R^2$ (~0.1) or $\varepsilon_{p\text{-}CO2}$ (maximum $R^2$ with $pCO_2$ (~0.38), despite high significance (low p-values) of the regressions. The explained variance increases when polynomial regressions are applied. Several controlled growth experiments indeed show a non-linear response of $\varepsilon_p$ as a function of $pCO_2$ of the growth medium (Hoins et al., 2015) although the number of data points in such experiments limit full mathematical descriptions of the trends within the $pCO_2$ range of this field study. Here, a second-order polynomial (quadratic) regression achieves an $R^2$ of ~0.74 and ~0.79 for the non-weighted and weighted versions, respectively.

It is conceivable that other environmental parameters also significantly contribute to $\varepsilon_p$ variability (Fig. 6). For example, $[PO_4^{3-}]$, $[NO_3]$, and $pCO_2$ contribute significantly to a (linear) multiple-regression model, which takes the form of $\varepsilon_{p\text{-}CO2} = c + xCO_2 + yPO_4 + zNO_3$, where c, x, y and z are numerical constants. The multiple regression model using these three parameters covers ~58% of the variance in *O. centrocarpum* $\varepsilon_{p\text{-}CO2}$ (not weighted), and 67% when weighted to number of measurements per sample. Including more parameters, such as SST, oxygen concentrations, or other carbonate system parameters, explains slightly more of the observed variance, but does not significantly improve the model. The residual mean standard error (RSME) of the CNP-$\varepsilon_p$ multiple regression model is ~1.45 ‰ while a linear regression with only $pCO_2$ yields 1.7 ‰. Only weighted regressions are given here and reported ranges of the constants represent one standard error or equivalent percentiles in case of Monte Carlo simulated errors. These models have the following optimal formats:

Equation 1 linear:

$\varepsilon_{p\text{-}CO2} = 6.6 \pm 2.1 + 0.031 \pm 0.008 \, pCO_2$

(Adjusted $R^2 = 0.48$, p = 0.001, RSME = 1.7 ‰) (Fig. 6B)

Equation 2a quadratic (data without error propagation and only suitable for use > 240 µatm)

$\varepsilon_{p\text{-}CO2} = 40.8 \pm 7.2 - 0.23 \pm 0.055 \, pCO_2 + 4.88 \times 10^{-4} \pm 1 \times 10^{-4} \, pCO_2{}^2$

(Adjusted $R^2 = 0.79$, p <0.001, RSME = 1.13 ‰) (Fig. 6B)

Equation 2b quadratic (Monte Carlo constrained errors – analytical for $pCO_2$ and $\varepsilon_{p\text{-}CO2}$) (Fig. 7B)

$\varepsilon_{p\text{-}CO2} = 35.6 \, {}^{+5.8}/_{-5.6} - 0.19 \, {}^{+0.045}/_{-0.045} \, pCO_2 + 4.1 \, {}^{+0.91}/_{-0.88} \, 10^{-4} \, pCO_2{}^2$

Equation 2c quadratic (as 2b with additional $45 \pm 15$ ppm $pCO_2$ error) (Fig. 7C)

$\varepsilon_{\text{p-CO2}} = 39.3\ ^{+11.5}/_{-8.8} - 0.19^{+0.058}/_{-0.076}\ pCO_2 + 3.4\ ^{+1.3}/_{-0.95}\ \text{x } 10^{-4}\ pCO_2\ ^2$

Equation 2d quadratic (as 2b with resampled $pCO_2$ rise 1800 – 2000 CE) (Fig. 7D)

$\varepsilon_{\text{p-CO2}} = 29.8\ ^{+11.0}/_{-8.0} - 0.13\ ^{+0.061}/_{-0.084}\ pCO_2 + 2.6\ ^{+1.5}/_{-1.1}\ \text{x } 10^{-4}\ pCO_2\ ^2$

Equation 3a multiple-regression linear (Fig. 6E):

$\varepsilon_{\text{p-CO2}} = 6.0 \pm 3.1 + 0.034 \pm 0.01\ pCO_2 + 1.22 \pm 0.47\ NO_3 - 10.85 \pm 3.7\ PO_4^{3-}.$

(Adjusted $R^2 = 0.67$, p $< 0.001$, RSME = 1.45 ‰)

Equation 3b adjusted for application in the paleo-domain:

$\varepsilon_{\text{p-CO2}} = 6.0 \pm 3.1 + 0.034 \pm 0.01\ pCO_2 - 1.1 \pm 5.3\ PO_4^{3-}.$

Equation 4 multiple-regression linear:

$\varepsilon_{\text{p-DIC}} = 18.4 \pm 3.1 + 0.025 \pm 0.01\ pCO_2 + 1.45 \pm 0.47\ NO_3 - 11.1 \pm 3.7\ PO_4^{3-}$

(Adjusted $R^2 = 0.52$, p $= 0.01$, RSME = 1.44 ‰)

The two linear multiple regression models are offset (Equations 3a and 4), primarily because of the carbon isotope fractionation between $HCO_3^-$ and $CO_2$. The slope with respect to $pCO_2$ also varies slightly between the models for $\varepsilon_{\text{p-DIC}}$ and $\varepsilon_{\text{p-CO2}}$ due to the temperature dependent fractionation between $HCO_3^-$ and $CO_2$, but the slopes with $NO_3^-$ and $PO_4^{3-}$ are indistinguishable. The quadratic regression seemingly better fits the variability observed in $\varepsilon_{\text{p-CO2}}$ compared to other (multiple) linear regressions and removes any structure in the residuals, potentially signaling a non-linear response in $\varepsilon_{\text{p-CO2}}$ to $pCO_2$. The quadratic

regression also indicates insensitivity to $pCO_2 \leq 240$ µatm and should not be used below this value (Fig. 6B). The Monte Carlo simulations of scenarios where an additional $pCO_2$ uncertainty is imposed as a normally distributed mixture of pre-industrial values and modern, offsetting $pCO_2$ by $+45 \pm 15$ ppm (Fig. 7C, Eq. 2c), and a resampled uncertainty derived from the $pCO_2$ rise since pre-industrial times (Fig. 7D, Eq. 2d) show that the parameters of the quadratic regression are fairly robust to these uncertainties (*i.e.* none of the parameters become insignificant at p $> 0.05$), although the absolute $pCO_2$ values and errors

increase.

## 4. Discussion

### 4.1 Absolute values, comparison to marine organic matter

The recorded $\delta^{13}C_{DINO}$ range and absolute values (~ -18‰ to -35‰) correspond well with global $\delta^{13}C$ values previously reported for marine particulate organic matter ($\delta^{13}C_{POC}$) (e.g. Freeman and Hayes, 1992; Goericke and Fry, 1994) and modelled phytoplankton biomass (e.g. Magozzi et al., 2017; Tagliabue and Bopp, 2008). Consequently, $\varepsilon_{p\text{-DIC}}$ and $\varepsilon_{p\text{–CO2}}$ are also within the expected range for general marine particulate organic matter. However, the intra-sample variance of $\delta^{13}C_{DINO}$ appears to be substantial, often spanning most of the full range (~10‰) observed for $\delta^{13}C_{POC}$. Some of the observed variability might be related to the limited analytical precision during measurements of the extremely small amounts of carbon of individual dinocysts. Fully constraining the contribution of this analytical uncertainty to the observed variance is, however, not possible because of unresolvable micrometer-scale heterogeneity in the PE standard (van Roij et al., 2017; Sluijs et al., 2018). In most cases, the variance in $\delta^{13}C_{DINO}$ is similar to that of the standard. Still, it is likely that some of the seasonal $\delta^{13}C_{DIC}$ differences are also recorded in the $\delta^{13}C_{DINO}$, and that some additional inter-specimen $\delta^{13}C$ variance is present. This is to be expected since the $\delta^{13}C$ populations from our integrated core-top samples span seasons, decades and thus also considerable variability in seawater properties and population change. In addition, growth-induced randomness and changes in $\delta^{13}C$ and DIC in the cell's microenvironment likely contributed to inter-specimen variability. Note that in our data inter-specimen variability is still underestimated because we analyzed 3-7 specimens per ablation event, as single-cyst carbon yield (~7 ng C) from these cyst-sizes approached the limit for reliable measurements (van Roij et al., 2016). We minimized potential influence of differences in cell size or shape through manual selection. We thus analyzed a population where the pre-selection of similar-sized cysts restricts the variance in cell surface area and volume, unlike biomarker-based proxies for which the cell size has to be reconstructed independently (Henderiks and Pagani, 2007; Stoll et al., 2019). This approach could reduce scatter in the relation of $\varepsilon_p$ to environmental variables (Popp et al., 1998).

### 4.2 Cell – cyst offset

One of the striking differences between the here generated data and the existing culture experiments, is that carbon isotope fractionation of dinocysts in the natural environment appears to be much larger than that of motile cells from controlled growth (dilute batch) experiments (Hoins et al., 2015, 2016b). We find average $\varepsilon_p$ values ranging between 13–20‰ and 23–29‰ with respect to CO₂ and DIC. Cultured cells of *O. centrocarpum* yielded not only a smaller overall $\varepsilon_p$, but also a smaller range (~8– 12 and 18.5–20‰) across a larger CO₂ gradient, implying the cysts have a much steeper fractionation slope with CO₂ compared to the motile cells. Despite these differences, the average $\varepsilon_p$ for *Spiniferites* species (*S. ramosus*, *elongatus* & *mirabilis*) is often somewhat larger than for *O. centrocarpum* (Fig. 4). This is consistent with culture experiments that showed larger CO₂-dependency and overall slightly larger $\varepsilon_p$ in the motile species *G. spinifera* compared to *P. reticulatum* (Hoins et al., 2015). While the cultured single strains and dinoflagellate populations in nature may behave somewhat differently, we do not expect that this alone underlies such a marked offset between the motile cultured cells and natural cysts. Natural cysts and cultured cells seem consistently offset in $\delta^{13}C$, although at present the exact amplitude of this offset cannot be determined. However, such an offset is in line with certain compounds in dinoflagellate cells being depleted in $^{13}C$ relative to the bulk biomass

(Schouten et al., 1998; Wilkes et al., 2017). The organic-walled dinocysts consist of mostly aliphatic and aromatic compounds, forming a complex biopolymer referred to as dinosporin (de Leeuw et al., 2006; Versteegh et al., 2007, 2012). Depending on the biosynthetic pathway of the cyst-material and the derivation or degradation of the original compounds, this may result in

offsets in $\delta^{13}C$ values between cysts and the motile cells. A potential additional fractionation might be introduced during taphonomy and also later by the processing of sediments to concentrate the dinocysts from sediment samples. The sediment processing involves hydrochloric and hydrofluoric acids, which affects the more labile organic compounds. Last, it is conceivable that fractionation in the dilute batch experiments may be reduced by e.g., higher-than-natural growth rates. This may be supported by chemostat culture experiments on *Alexandrium tamarense* (Wilkes et al., 2017), which show a (much)

greater fractionation compared to the dilute batch experiments (Hoins et al., 2015). However, the enhanced fractionation recorded in chemostat experiments is likely an artifact of isotope equilibration times exceeding $CO_2$ uptake rates (Brandenburg et al., 2022; Zhang et al., 2022). The range of options cannot be narrowed down until cultured cysts are compared to their motile cells harvested from the same culture, and treated with similar techniques as used for the sediments. Until these data become available, inferences on the origin and amplitude of the offsets between the cells and cysts of *O. centrocarpum* and

*Spiniferites* remain speculative.

### 4.3 Environmental controls on carbon isotope fractionation

Carbon isotope fractionation is determined by RubisCO and several environmental parameters, dominantly $pCO_2$, but also cell size and shape, growth rates and nutrient or light regimes (e.g. Freeman and Hayes, 1992; Pagani, 2013; Popp et al., 1998;

Stoll et al., 2019 and many others). In most cases, fractionation is $pCO_2$ dependent, and consequently $\varepsilon_p$ in various groups and genera has been used as a paleo-$pCO_2$ proxy (e.g., Freeman and Hayes, 1992; Pagani, 2013; Rae et al., 2021). We performed a broad-spectrum multiple regression analysis to identify environmental factors that contribute significantly to dinocyst $\varepsilon_p$. The most important parameter is $pCO_2$, in line with previous studies (Freeman and Hayes, 1992). A large part of the remaining variability can be attributed to growth rate and ultimately nutrient content, specifically nitrate and phosphate of the surface

waters, in line with previous studies on phytoplankton and dinoflagellate carbon isotope fractionation (Popp et al., 1998; Hoins et al., 2016b; Wilkes et al., 2017; Wilkes and Pearson, 2019). We find that $pCO_2$ and $\varepsilon_p$ are positively correlated, while $NO_3^-$ and $PO_4^{3-}$ are negatively correlated with $\varepsilon_p$ (Fig. 6C,D).

The inclusion of nutrient levels as environmental factors on the carbon isotope fractionation reduces the error in the fit between the measured and modelled fractionation. In the rare cases where paleo-nutrient concentrations are estimated, reconstructions

usually cover only $[PO_4^{3-}]$, implying the $[NO_3^-]$ in Eq. 3a and 4 has to be approximated from $[PO_4^{3-}]$. As $[PO_4^{3-}]$ and $[NO_3]$ are generally well-correlated (typically ~1:10 $[PO_4^{3-}]:[NO_3]$; here ~1:8) in ocean waters, this is relatively straightforward. For application in the paleo-domain however, nutrient ($[PO_4^{3-}]$) reconstructions may not be available or considered too unreliable to provide meaningful constraints. In specific cases where accurate paleo-$[PO_4]$ estimates or large changes are reconstructed, Eq. 3b may be applied, in which $[NO_3]$ is substituted for $[PO_4^{3-}]$ in a 1:8 ratio. However, unless there are clear changes in the

nutrient regime or sufficient proxy constraints on the nutrient concentrations, we prefer a calibration based exclusively on carbon isotope fractionation and carbonate system parameters, including sea surface temperature to calculate $\delta^{13}C_{CO_2}$ from $\delta^{13}C_{DIC}$, to reconstruct $p$CO$_2$.

### 4.4 Influence of carbon concentrating mechanisms: CO$_2$ insensitivity

As many phototrophs, including dinoflagellates, have mechanisms for concentrating CO$_2$ near the cell membrane, the sensitivity of carbon isotope fractionation to ambient CO$_2$ is expected to diminish below a certain concentration (M.R. Badger, 2003, M.P.S. Badger, 2021; Stoll et al., 2019). This is particularly the case as dinoflagellates utilize type II RubisCO, which has generally poorer performance compared to type I RubisCO at low CO$_2$ concentrations (Giordano et al., 2005). Indeed, substantial activity of the carboxyl anhydrase (CA) enzyme, which facilitates the conversion of HCO$_3^-$ to CO$_2$ inside the cell

or near the membrane, was observed in numerous dinoflagellate species, including *Lingulodinium* (Lapointe et al., 2008), *Symbiodinium* (Leggat et al., 1999) and the here analysed *Operculodinium* (Ratti et al., 2007). Also here we find a relatively low sensitivity at the lower end of the CO$_2$ scale. The lower CO$_2$ values correspond to the northern-most locations, with trends below 240 µatm becoming somewhat obscured, at a minimum $\varepsilon_p$-CO$_2$ around 13 ‰ (Fig. 6B). However, part of this levelling of the proxy-relationship may reflect the locally higher nutrient concentrations offsetting the higher CO$_2$. Though growth rates

have a clear influence on $\varepsilon_p$ in algal groups (Burkhardt et al., 1999), including dinoflagellates (Wilkes et al., 2017), the dilute batch culturing experiments conducted with *P. reticulatum* showed no clear influence of growth rates on $\varepsilon_p$ (see also Sec. 4.2). It is also conceivable that higher growth rates influence $\varepsilon_p$ indirectly through, for example, seasonally enhanced CO$_2$ drawdown, resulting in higher $\delta^{13}$C values in the remaining DIC. This effect may be enhanced by the relatively short growing season at the high latitudes. However, in culture experiments at low CO$_2$ concentrations with other dinoflagellate species, $^{13}$C-

fractionation was higher under nutrient limiting conditions than under replete conditions (Hoins et al., 2016b). Because of these confounding factors, the influence of carbon concentrating mechanisms on $\varepsilon_p$ in *O. centrocarpum* is difficult to gauge with the presently available data, and would ideally be tested using high nutrient or very low CO$_2$ concentrations.

Still, also in the relatively limited range the current ocean offers for testing $p$CO$_2$ proxies we have established a robust, albeit not overly sensitive, relation between $p$CO$_2$ and dinocyst $\delta^{13}$C. Our cyst-based calibration yields more conservative and

arguably more realistic absolute CO$_2$ estimates and variability compared to available culture-based calibrations as it is based on the same compounds as will be analyzed in the paleo-domain. However, the low sensitivity at low CO$_2$ implies that, until better constraints become available, the proposed calibration is potentially less suitable for application across, for example, the Pleistocene glacial periods. Further, it is important to realise that the value of 240 µatm is based on the assumption that the $\varepsilon_p$-CO$_2$ – $p$CO$_2$ relation originated from cysts that have not been affected by the Suess-effect and thus represent a lower limit for

CO$_2$ (in)sensitivity. While our data does not preclude fractionation smaller than the here observed minimum (~13 ‰) during low $p$CO$_2$ periods, increased sensitivity at higher CO$_2$ suggests that CO$_2$ above (minimum) 240 µatm and CO$_2$ variability can be reconstructed with reasonable precision.

#### 4.5 Challenges of age-control and potential caveats associated with anthropogenic carbon

A topic that warrants specific attention is the potential impact of anthropogenic carbon emissions on shaping the relation between $\delta^{13}C_{DINO}$ and $\varepsilon_p$. Here, we assume that our outlier analyses preferentially excluded samples that were significantly affected by anthropogenic $CO_2$ and that, on average, the remainder of our core-top $\delta^{13}C_{DINO}$ were not appreciably affected by either decreasing $\delta^{13}C_{DIC}$ or elevated $pCO_2$. The similarity in pre-industrial down-core $\delta^{13}C_{DINO}$ and that of three core-top localities in the North Atlantic corroborate the validity of this assumption locally but these observations cannot be extrapolated

to other regions. Unfortunately, sedimentation rates or other constraints for cyst production datums are not available. Further, if sedimentation rates were available for core-top localities, that would constitute an imperfect solution as it cannot provide the required dinocyst-specific age distribution needed to obtain an appropriate local $pCO_2$ correction to the datum of cyst production. This challenge may be unique to data such as presented here, as studies on other substrates with $pCO_2$-proxy potential either could not generate data for individual single celled organisms or have avoided the issue through other means,

such as culture experiments (e.g. Pagani et al., 2002; Henehan et al., 2013), approaches that have other drawbacks.

Although we have no constraints on the ages of the cysts analysed here, we can provide a meaningful test of the potential uncertainty added by our assumption that cysts are representative of pre-industrial conditions (Eq. 2a, b). The two scenarios that we have explored through Monte-Carlo simulations show that, depending on the error distribution imposed on the assumed $pCO_2$ used in the quadratic regressions (Eq. 2a,b), the fitted regression shifts towards higher values (Fig. 7C) and may be

steeper (Fig. 7D). These error-propagation case studies illustrate that our proposed pre-industrial transfer function (Eq. 2a, b), if it indeed contains a substantial proportion of very recent dinocysts, is likely to lead to underestimated $pCO_2$ and perhaps $pCO_2$ variability when applied in the paleo-domain. Consequently, we recommend future studies target, for example, sediment trap collections and culture-derived dinocysts to validate the results obtained here.

### 5. Conclusions - Proxy potential, limitations and calibrations

Our new modern ocean single-species carbon isotope fractionation dataset shows promising trends with environmental variables, $pCO_2$ and nutrients. The selection of individual cysts allows control of cell size and species, which reduces uncertainty in proxy calibration and application compared to approaches based on organic substrates which inevitably integrate entire communities. Although this approach has clear benefits, it also poses a unique challenge as the impact of anthropogenic

carbon emissions on individual single celled organisms must be considered. Based on our analyses, we expect this to be a relatively minor factor. In a worst-case scenario, however, we find that, although a helpful simplification, the assumption that all dinocysts from the core-top samples represent pre-industrial conditions may lead to an underestimate of $pCO_2$ and perhaps also $pCO_2$ variability when applied as a proxy in the paleo-domain.

In addition, many of the challenges associated with other proxies based on organic substrates are encountered here as well. For

example, like in cultures (Hoins et al., 2016b), we observed an impact of nutrients on carbon isotope fractionation, possibly related to differences in growth rates. Similarly, at low $pCO_2$ values sensitivity is reduced, possibly because of carbon

concentrating mechanisms involved in dinoflagellate C uptake, as observed in culturing experiments (e.g., Hoins et al., 2016a). Another remaining challenge is the observed difference between the cultured populations and cysts from the core top sediments. This is a pronounced difference, not only in the absolute isotope fractionation values but also in the slope of the $CO_2$ sensitivity, which appears to be much larger for the cysts and requires attention in future culture studies.

The offset in $\delta^{13}C$, combined with uncertainties in fractionation between the motile cells and dinocysts imply that $CO_2$ reconstructions using culture-based calibrations are more likely to overestimate past $pCO_2$. Furthermore, the large spread in our data (~5‰ between high and low $CO_2$) suggests that, due to this high sensitivity in the cysts, the method is also suited to study population dynamics.

**Data availability**

All newly generated data will be available via a permanent online repository (Mendeley data doi: 10.17632/z6285myxkm.2) upon publication.

**Author contribution**

AS & GJR designed the study, LvR, IK & JF processed samples, generated and analysed data, JF wrote the original draft, AS & GJR reviewed and edited the manuscript. AS acquired funding for this study.

**Competing interests**

The authors declare that they have no conflict of interest.

**Acknowledgments**

We thank A. van Dijk, M. Kienhuis and H. de Waard (Utrecht University) for technical and analytical assistance. AS acknowledges funding from Netherlands Organisation for Scientific Research (NWO) #ALWOP.223 and European Research Council (ERC) Starting grant #259627. This work was carried out under the program of the Netherlands Earth System Science Centre (NESSC), financially supported by the Dutch Ministry of Education, Culture and Science. We thank two anonymous reviewers for their insightful and highly constructive comments and Steven Bouillon for editorial handling.

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

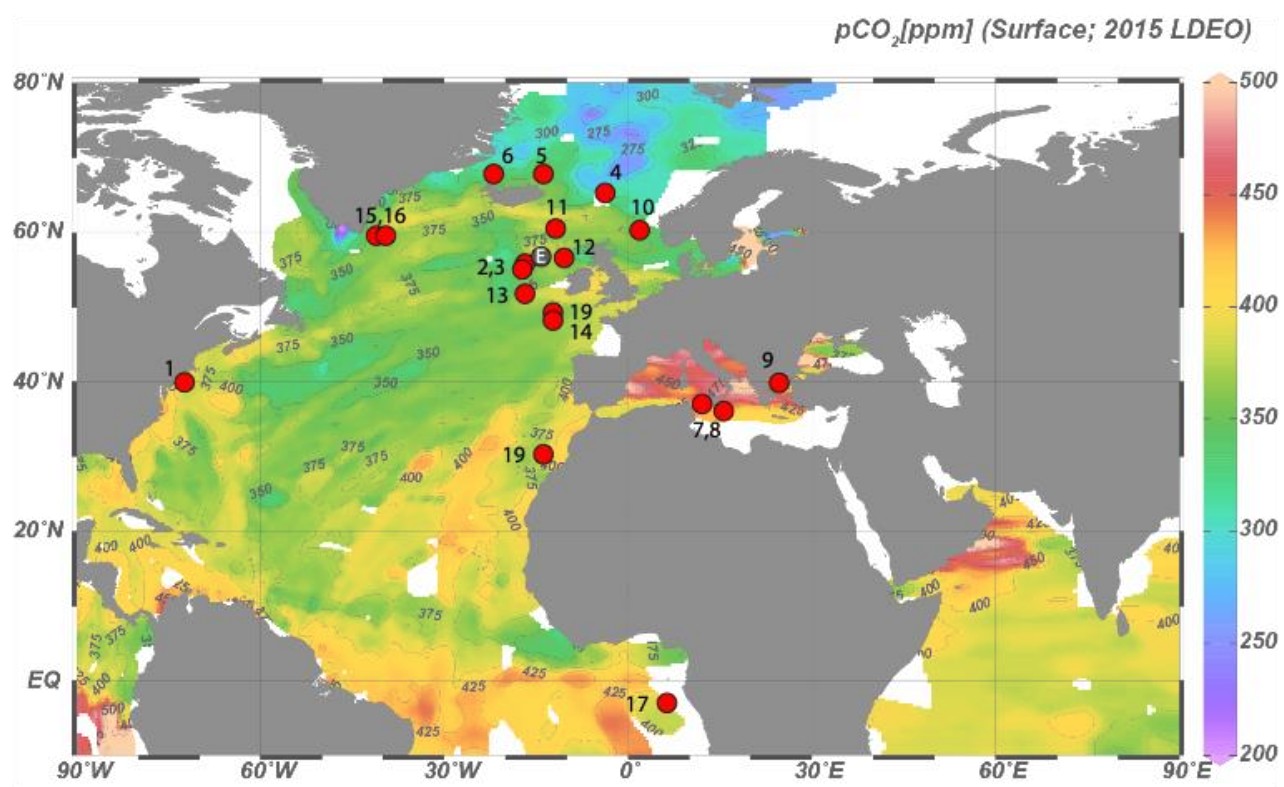

**Figure 1.** Locations of samples with sufficient *Operculodinium centrocarpum* and/or *Spiniferites* spp for dinoflagellate cyst δ[13]C analyses. Numbers correspond to localities listed in Table 1. Down-core data location (ENAM9606; 55.650 ºN, -13.985 ºE) is marked by a grey dot ("E") between core-tops #2, 3 and 12.


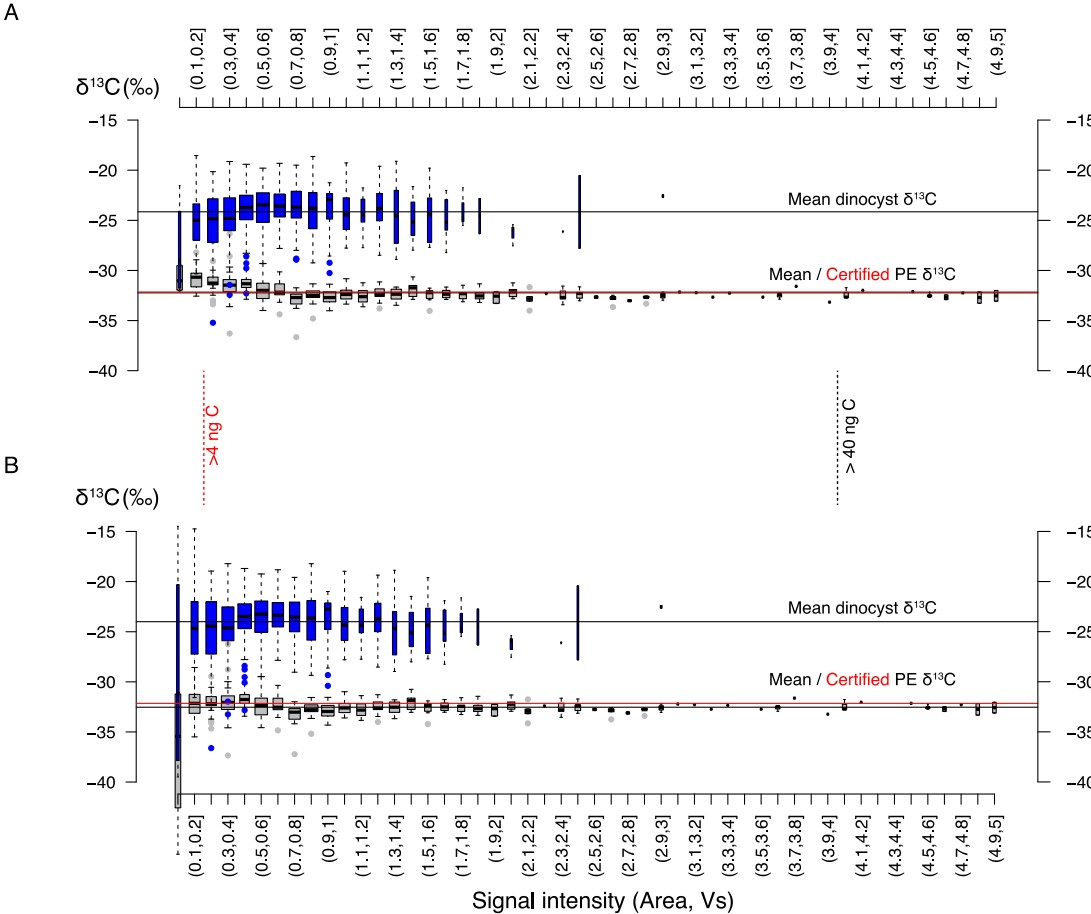

**Figure 2. Signal intensity in Volt seconds (Vs) versus carbon isotope value distribution within bins of 0.1 Vs. A.** The blue boxplots represent our new dinocyst $\delta^{13}C$, and the gray boxplots represent the previously published data for the PE standard ($\delta^{13}C$ value −32.15‰ ± 0.05 ‰; 1σ). **B.** Same as A. but for background-corrected values (see §3.2). The vertical black and red line represent cut-off values for individual PE standard measurements (>4 Vs; ~42 ng C; 0.5‰ precision) and individual cyst measurements (>0.2 Vs; ~6 ng C), respectively. The width of each boxplot is square-root scaled with the number of measurements in the respective bins. Note that several bins at the high-end do not contain any data.

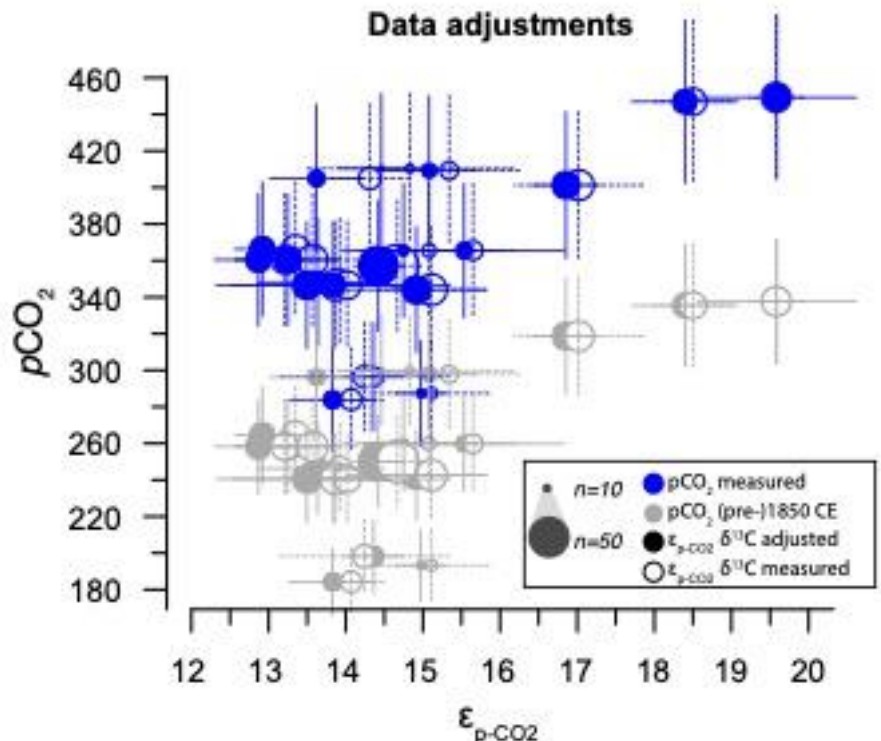


**Figure 3. Effects of δ¹³C and pCO₂ corrections (see also Figure 2 and §3.2).**

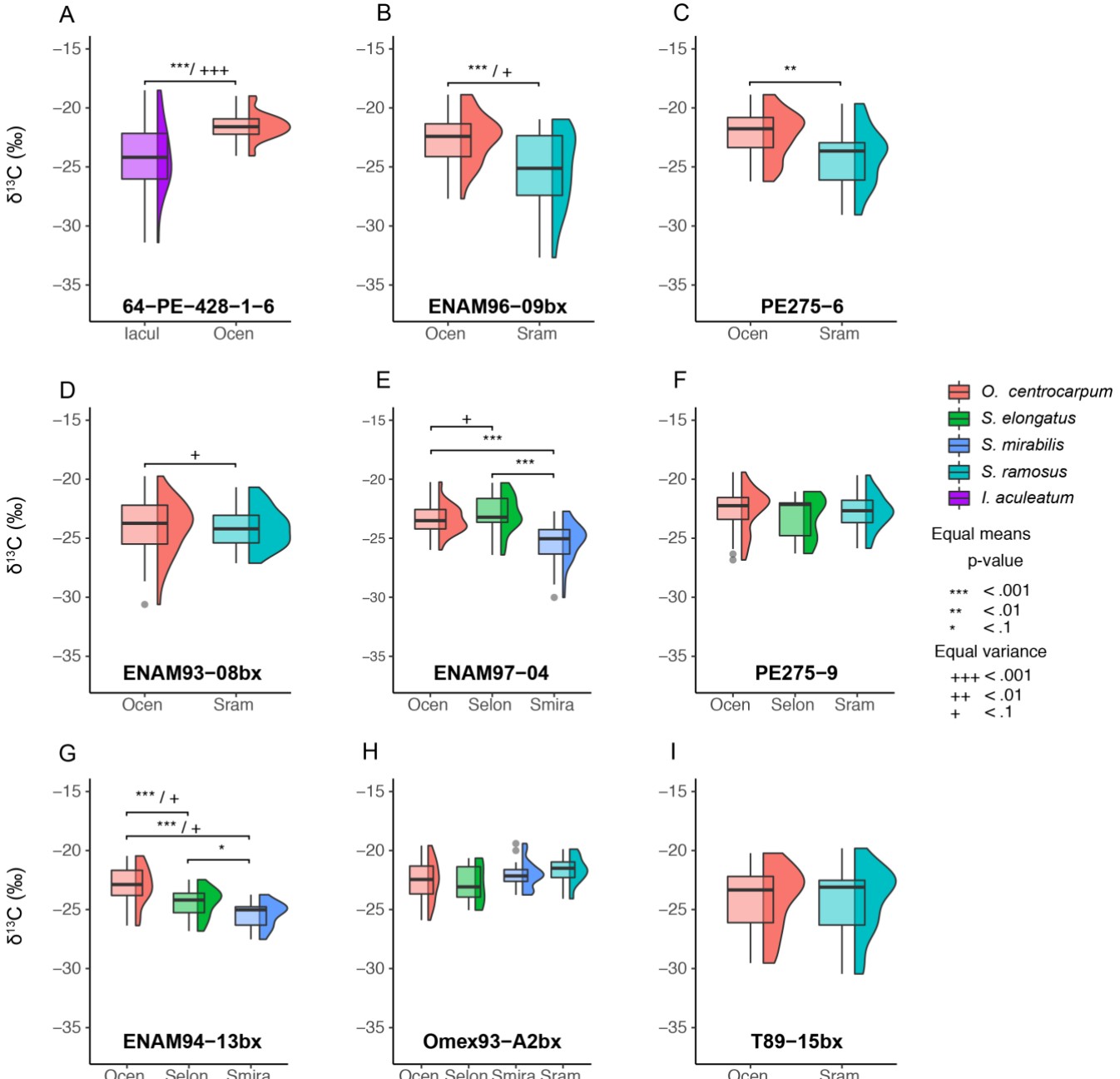

**Figure 4.** Carbon isotope measurements for multiple species; each panel represents a single sample after eliminating extremely small measurements sizes (<0.2 Vs), background correction and removing outliers (± 2.5 interquartile range) (paragraph §3.2). Each box-whisker and $\delta^{13}C$ distribution plot represents a set of measurements for a single species at their respective locality; note that tailing towards negative $\delta^{13}C$ is common. Brackets above species $\delta^{13}C$ populations indicate significant differences in means and variance for different species within a single sample.

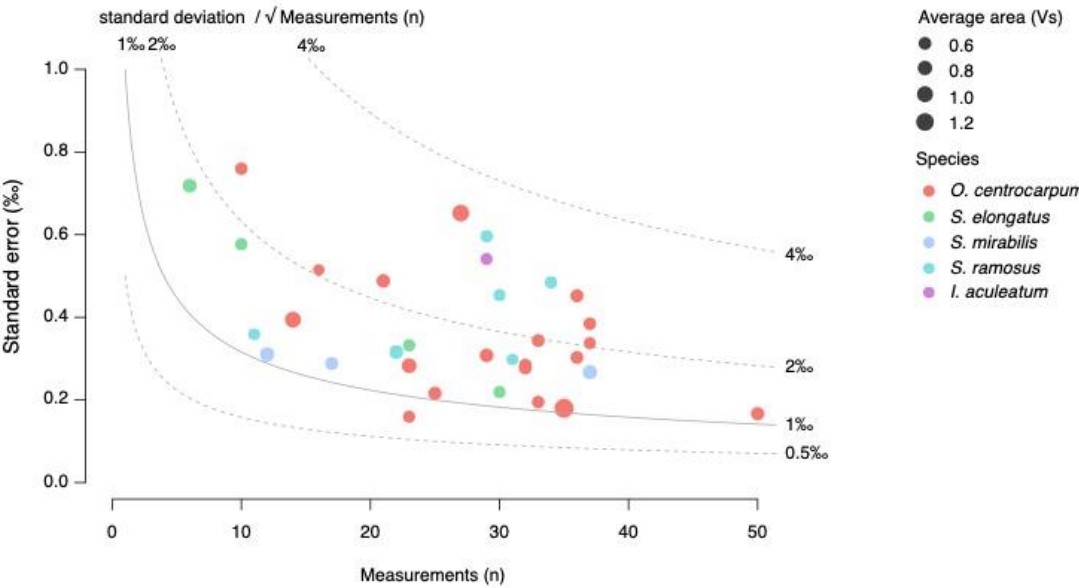

**Figure 5.** Relation of standard error of $\delta^{13}C_{DINO}$ (‰) with the number of measurements and signal intensity (area in Volt seconds (Vs)). Colors correspond to the various analyzed species.

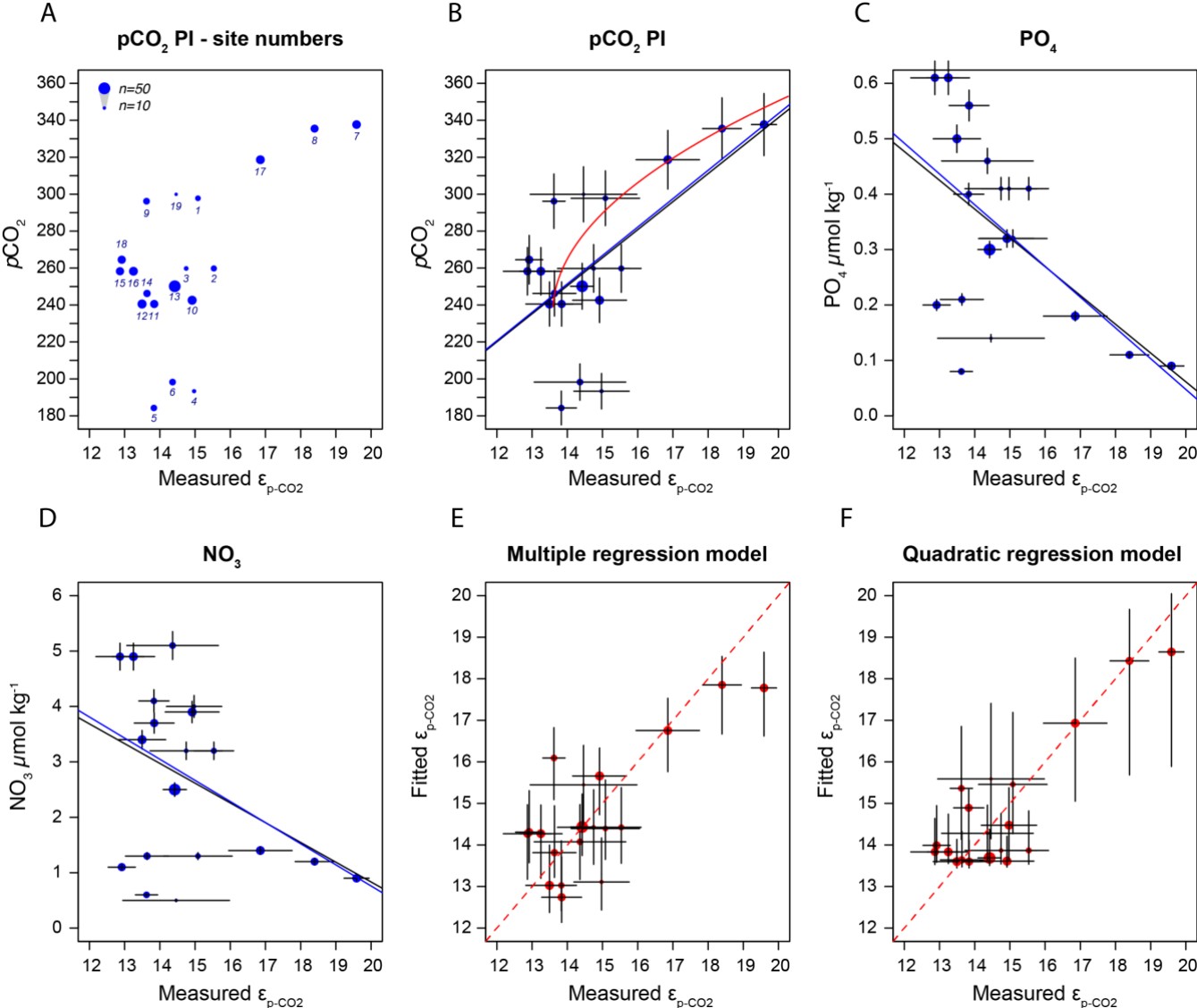

**Figure 6. A.** Sample – specific ε_p-CO2 of *O. centrocarpum* relative to pre-industrial pCO2, numbers correspond to localities listed in Table 1. **B.** Regression analyses for ε_p-CO2 of *O. centrocarpum* relative to $pCO_2$ (measured, corrected to pre-industrial values); black line represents simple linear regression, blue lines represent weighted linear regressions and red line represents weighted quadratic regression. **C.** phosphate concentrations ($PO_4^{3-}$), **D.** nitrate concentrations ($NO_3^-$). **E.** Fitted values illustrating the multiple regression model performance using parameters a-c relative to measured ε_p-CO2. **F.** Fitted values using only pre-industrial $pCO_2$ but applying a quadratic regression (red curve in panel b). Errors in panels b-d represent 5% of the measured value and errors on the fitted values in panels d and e represent propagated errors of both measurements and environmental variables (as shown in panel B – D) using Monte-Carlo simulations ($n = 1000$) for regression models. Symbol size (top left corner of panel A) represents the number of measurements within each sample.

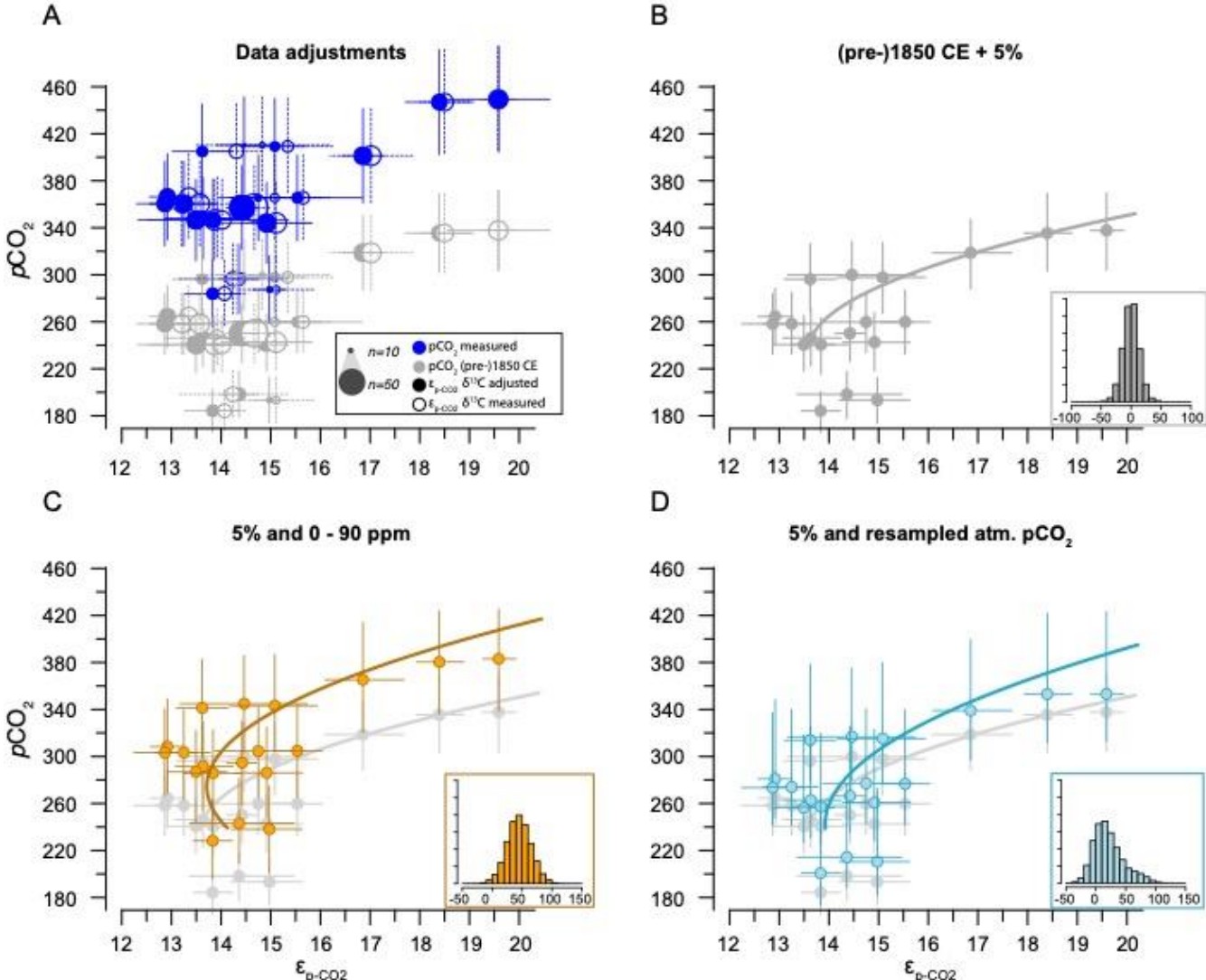

**Figure 7. Sensitivity tests to potential effect of anthropogenic carbon emissions. A.** Effects of data treatment on the difference between measured and adjusted $pCO_2$ and $\varepsilon_{p\text{-}CO2}$ (same as Figure 3). Open symbols indicate measured $\delta^{13}C$, closed symbols represent data after eliminating small signals (<0.2 Vs) and outliers. Blue dots represent measured $CO_2$ values and grey dots indicate the $CO_2$ around 1850 CE. **B.** Quadratic regression (red line in Fig. 6B) with propagated analytical error on $pCO_2$ and $\delta^{13}C$ only, using $CO_2$ values around 1850 CE (grey filled symbols in panel A. **C.** As in B but with addition of a $+45 \pm 15$ ppm error to reflect potential impact of anthropogenic $CO_2$ in orange. Grey dots and curve of panel B are added as a comparison. **D.** As in B but with addition of a $CO_2$ increase relative to pre-industrial, sampled for the period $1800 - 2000$ CE. Insets on the bottom right in panels B, C and D show the combined error distributions imposed on $pCO_2$. All error bars in panels B – D on $pCO_2$ and $\varepsilon_{p\text{-}CO2}$ are $2.5 - 97.5\%$ percentile ranges from Monte Carlo simulations (n = 1000).

| Site number | Core-ID | Latitude (ºN) | Longitude (ºE) | Species | Measurements (n) | S-W normality |
|---|---|---|---|---|---|---|
| 1 | NF2012-091 | 37.977402 | -73.669403 | *O.centrocarpum* | 22 (21) | * () |
| 2 | PE360-24 | 55.496231 | -15.800755 | *O.centrocarpum* | 24 (23) | ** () |
| 3 | PE360-45 | 55.539398 | -15.8453 | *O.centrocarpum* | 23 (16) | () |
| 4 | NA87-02 | 64.480003 | -5.83 | *O.centrocarpum* | 20 (14) | *** () |
| 5 | LCD13 | 67.501282 | -15.069252 | *O.centrocarpum* | 25 (25) | () |
| 6 | LCD10A | 66.677437 | -24.179598 | *O.centrocarpum* | 29 (27) | () |
| 7 | MedSea (MC-613) | 35.8575 | 14.105556 | *O.centrocarpum* | 35 (35) | () |
| 8 | MedSea (MC-614) | 35.8075 | 12.998056 | *O.centrocarpum* | 33 (32) | * () |
| 9 | MedSea (MC-645) | 40.2175 | 25.244167 | *O.centrocarpum* | 33 (23) | ** () |
| 10 | ENAM93-08bx | 59.501667 | 3.69 | *O.centrocarpum* | 38 (37) | *() |
| | | | | *S.ramosus* | 33 (32) | () |
| 11 | ENAM94-13bx | 60.249997 | -11.19 | *O.centrocarpum* | 36 (33) | () |
| | | | | *S.elongatus* | 34 (30) | *** () |
| | | | | *S.mirabilis* | 13 (12) | () |
| 12 | ENAM96-09bx | 57.159917 | -10.26 | *O.centrocarpum* | 39 (37) | () |
| | | | | *S.ramosus* | 30 (29) | () |
| 13 | ENAM97-04 | 52.410386 | -14.94 | *O.centrocarpum* | 52 (50) | () |
| | | | | *S.elongatus* | 23 (23) | () |
| | | | | *S.mirabilis* | 38 (37) | ** (*) |
| 14 | Omex93-A2 bx | 49.483 | -11.13 | *O.centrocarpum* | 30 (29) | () |
| | | | | *S.ramosus* | 13 (11) | *() |
| | | | | *S.elongatus* | 6 (6) | () |
| | | | | *S.mirabilis* | 18 (17) | () |
| 15 | PE275-6 | 59.272369 | -38.36 | *O.centrocarpum* | 34 (33) | *() |
| | | | | *S.ramosus* | 34 (30) | **() |
| 16 | PE275-9 | 59.272369 | -38.36 | *O.centrocarpum* | 37 (36) | *(*) |
| | | | | *S.ramosus* | 23 (22) | *(*) |
| | | | | *S.elongatus* | 12 (10) | *(*) |
| 17 | T89-15bx | -4.199372 | 10.05 | *O.centrocarpum* | 36 (36) | * (*) |
| | | | | *S.ramosus* | 35 (34) | * (*) |
| 18 | 64PE428-1-1-6 | 47.079782 | -10.197305 | *O.centrocarpum* | 35 (33) | () |
| | | | | *I.aculeatum* | 30 (29) | () |
| 19 | 64PE428-1-6-6 | 30.67917 | -11.930478 | *O.centrocarpum* | 11 (10) | () |

**Table 1. Core localities, analyzed species, number of measurements and normality of the carbon isotope data.** Number of measurements total and in parentheses measurements used for environmental comparisons (see also results §3.4). Shapiro-Wilk (S-W) normality test on data: non-normal data distributions are indicated where *p* values are < 0.1 (*), <0.01 (**) and

<0.001 (***), in parentheses the same for the data used for environmental comparisons. Site numbers correspond to those in Figs. 1 and 6A.

**Table 2**: Linear regression coefficients and significance for all samples where *O. centrocarpum* was analyzed (n = 19), with $\varepsilon_{p\text{-}CO2}$ as dependent variable. Parameters with p-values <0.05 in bold.

| | Coeff. | Std.err. | t | p | R^2 |
|---|---|---|---|---|---|
| $CO_2$(mol/kg) | -2.506e+05 | 5.603e+05 | -0.447 | 0.66039 | 0.01163 |
| $CO_3^{2-}$(mol/kg) | 41263.663 | 15432.856 | 2.674 | 0.0160 | 0.296 |
| $HCO_3^-$(mol/kg) | 809.88 | 7057.32 | 0.115 | 0.910 | 0.0007741 |
| DIC(mol/kg) | 5655.859 | 6266.998 | 0.902 | 0.379 | 0.04572 |
| SST (°C) | 0.16971 | 0.06043 | 2.809 | 0.0121 | 0.3169 |
| SSS (psu) | 0.6737 | 0.3604 | 1.869 | 0.0789 | 0.1705 |
| $PO_4^{3-}$ (µmol /kg) | -5.5131 | 2.1102 | -2.613 | 0.0182 | 0.2865 |
| $NO_3^-$ (µmol /kg) | -0.4484 | 0.2493 | -1.798 | 0.0899 | 0.1599 |
| Si (µmol /kg) | -0.2553 | 0.3298 | -0.774 | 0.449 | 0.03405 |
| $O_2$ (mL/L) | -1.274 | 0.433 | -2.943 | 0.0091 | 0.3375 |
| ALK (mol/kg) | 7621.253 | 4712.178 | 1.617 | 0.124 | 0.1334 |
| $pCO_2$ ~1850 | 0.024945 | 0.007752 | 3.218 | 0.005050 | 0.3785 |

**Table 3**: As Table 2, but with $\varepsilon_{p\text{-}DIC}$ as dependent variable.

| | Coeff. | Std.err. | t | p | R^2 |
|---|---|---|---|---|---|
| $CO_2$(mol/kg) | 1.133e+05 | 4.760e+05 | 0.238 | 0.815 | 0.00332 |
| $CO_3^{2-}$(mol/kg) | 19589.987 | 14818.493 | 1.322 | 0.204 | 0.09322 |
| $HCO_3^-$(mol/kg) | 6547.21 | 5758.02 | 1.137 | 0.271 | 0.07068 |
| DIC(mol/kg) | 7803.835 | 5086.841 | 1.534 | 0.143 | 0.1216 |
| SST (°C) | 0.04959 | 0.06068 | 0.817 | 0.425 | 0.03781 |
| SSS (psu) | 0.4043 | 0.3201 | 1.263 | 0.224 | 0.08579 |
| $PO_4^{3-}$ (µmol /kg) | -2.3993 | 2.0318 | -1.181 | 0.254 | 0.07581 |
| $NO_3^-$ (µmol /kg) | -0.05666 | 0.22969 | -0.247 | 0.808 | 0.003567 |
| Si (µmol /kg) | -0.01925 | 0.28388 | -0.068 | 0.947 | 0.0002703 |
| $O_2$ (mL/L) | -0.4170 | 0.4385 | -0.951 | 0.355 | 0.05051 |
| ALK (mol/kg) | 6754.843 | 3956.586 | 1.707 | 0.106 | 0.1464 |
| $pCO_2$ ~1850 | 0.010083 | 0.007952 | 1.268 | 0.222 | 0.0864 |
