# Peer review of "Single-species dinoflagellate cyst carbon isotope fractionation in coretop sediments: environmental controls, CO2-dependency and proxy potential"

_Biogeosciences, 2022_

## Author Comment (AC1)

General Comments
Frieling et al., present records of carbon isotope fractionation from the resting cysts of dinoflagellates to investigate their utility in reconstructing ancient atmospheric CO2. This record of core-top material advances earlier work based on laboratory cultures (and based on sound theoretical basis) and so brings the community closer to confidence that this proxy may work in environmental settings. They show there are differences in carbon isotope fractionation between different species, emphasising the importance of single-species records, and show greater 13-C depletion in their core-top samples compared to cultured, motile organisms. The paper is interesting and makes an important contribution, but some of the analysis is unsatisfactory due to uncertainty about the age of the individual cysts in the "core-top" samples (detailed below). Therefore without a thorough treatment of that uncertainty (which is currently lacking) it's difficult to know whether this proxy has utility. There are certainly hints that it does, but unfortunately this paper does not yet demonstrate that compellingly.

**Author response:**
We thank the reviewer for recognizing the potential importance of our work and the constructive criticism. In the response below and in our revised manuscript we will further elaborate on (1) how the carbon isotope data from individual cysts has been treated and (2) further elaborate on the uncertainty in the age dating of the core-tops.

Specific Comments
The problem with using core-top samples is the substantial increase in atmospheric CO2 since the industrial revolution. As the authors note, it is highly uncertain whether the cysts are from the last week, the last year, decades or even centuries ago. The uncertainty around the contemporaneous CO2 is potentially very large. The "rough correction" to 1850 isn't really a correction at all, but an assumption which is not well supported, at best highly uncertain, and not really dealt with satisfactorily in the later analysis. The best approach (although expensive) would be to 14-C date some of these samples to see when this material actually dates to. The cheaper, and for this present study, more plausible approach would be to propagate through what is a really quite large uncertainty and see whether the conclusions still hold. Lines 147-8 state that "With the exception of pCO2, we hence assume all parameters (SST, SSS, nutrients) to be constant over the period the core top samples represent." A fundamental problem here is that the authors have little information (or at least present little data) about how long a period of time the core top samples do in fact represent. I'm not sure that the approach taken to this, systematically removing the most 13-C depleted samples is appropriate. Whilst it is certainly plausible that these individuals represent modern samples, the evidence is fairly circumstantial, and they could represent another confounding variable. What is the impact on the analysis if these samples are not removed?

**Author response:**
The reviewer comments on the potential of age-mixing of individual cysts in core-tops. This is very much a valid concern as we acknowledge in our original manuscript (lines 196-201). The main challenge is that core-tops (the top-most 2 cm of sediment) contain a range of ages of sedimentary components. While it might be possible to $^{14}$C-date carbonate or organic matter also these materials will derive from different times in the past and a single measurement will not show the range of ages of individual components such as for our dinocysts (i.e. the age-distribution of our individual dinocysts). Ideally one would date single-cell using $^{14}$C analyses but that is technically not feasible.
As the reviewer correctly points out, it is important to show the potential impact of our data-treatment and we include these analyses in a supplementary file to the revised manuscript. In

general, the impact of removing analyses based on exceptionally low amounts of C and exclusion of outliers has no appreciable impact on the regression parameters. The largest uncertainty comes from the recent addition of anthropogenic carbon, the Suess effect. Comparing our calibration including and excluding the Suess effect is to our opinion therefore the best approach to estimate the maximum uncertainty in the regressions.

[Figure]

*We will add the figure above to our supplementary information. The figure shows the difference between measured and adjusted $pCO_2$ and $\delta^{13}C$ ($\varepsilon p$-$CO_2$). Open symbols indicate measured $\delta^{13}C$, closed symbols represent data after eliminating small signals (<0.2 Vs) and outliers. Blue dots represent measured $CO_2$ values and red dots indicate the $CO_2$ around 1850 CE. For each dataset a simple linear regression, weighted to the number of measurements, is given. Dashed lines utilize measured $\delta^{13}C$, solid lines are from final $\delta^{13}C$ data. The red solid line is used in Figure 5A.*

The figure above illustrates the maximum error that may result from the Suess-effect by constructing a calibration with uncorrected data for each of the parameters (measured $\delta^{13}C$ and $CO_2$) and we compare this to the original calibration (solid red line). The main difference is the intercept of the calibration which is offset by the difference in atmospheric $pCO_2$ between 1850 and 2000 CE (*ca.* 100 µatm) resulting from adjustment of measured $CO_2$ levels to pre-industrial times. The slopes of all combinations are statistically indistinguishable. Further, we must stress the difference between the intercepts of the calibrations is a worst-case scenario and unlikely an accurate reflection of true uncertainty caused by the Suess-effect.

We also fully agree with the reviewer it is critical to show that the data-treatment is free of potential preconceived biases regarding which values to include or exclude. We will therefore further clarify in our revised manuscript how we first corrected for instrument drift at low C. This correction is exclusively based on repeat measurements of the IAEA-PE ($\delta^{13}C$ certified -32.15 ± 0.05‰) standard which show a convergence towards ~ -27‰ at the lowest intensities

(original manuscript lines 172-174; see also Van Roij et al., 2015). After performing the drift correction, we subsequently identify outliers within the species-specific populations measured on a single sample. We find a greater proportion of negative outliers (original manuscript line 215) compared to positive outliers (hence a skewed data set; original manuscript Table 1). Since we correct using the PE standard and exclude both negative **and** positive outliers there is no assumption that the most negative are the most recent cysts (which one might assume based on the Suess effect but would be an *a priori* interpretation).

The reason why we argue our data-treatment has removed most of the temporal bias that may exist between samples is as follows:
If we assume that (a) statistical distributions of $\delta^{13}C$ cyst populations are dominated by both $pCO_2$ and $\delta^{13}C-CO_2$ trends and that (b) these populations include a portion of both pre-industrial and recent times – the trends in both $pCO_2$ (higher $pCO_2$ > higher $\epsilon p$ > lower $\delta^{13}Ccyst$) and $\delta^{13}C-CO_2$ (lower $\delta^{13}C-CO_2$ > lower $\delta^{13}Ccyst$) would impose a (negative) skew on the $\delta^{13}Ccyst$ populations. Prior to correction for skewing at low intensities and outlier removal some, but not all, samples show a negative skew, while after outlier-removal the vast majority is statistically indistinguishable from a normal distribution (Table 1 in the original manuscript). In addition to the supplementary file showing the minor impact of excluding outliers (see above), we will clarify this rationale in our revised manuscript.

Ie L 213-4 "We assume these assemblages are representative of ocean conditions prior to the massive increase in anthropogenic carbon emissions." This is very sizable assumption which whilst plausible is not currently supported by very much evidence.

**Author response:**
Please also see response above – we now supply a more elaborate reasoning why we assume the vast majority of the analysed cyst populations is considered to represent pre-industrial times. We will clarify in our revised manuscript that this is also based on the assumption that core-top material includes an age range, likely centuries up to millennia. For core-top samples a major impact of the Suess effect may be detectable only in the relatively few cysts that were formed in the past 50 years.

It appears from Figure 5a that no uncertainty at all has been applied to the assumed CO2 value – is this correct? This is not a fair assumption given the uncertain age of each cyst. In fact a plain reading of Figure 5a suggests that, rather than supporting a function between CO2 and ep apart from at <240 uatm CO2, ep is effectively constant, and only slightly higher above 310. Why therefore has 240 uatm been emphasised?

**Author response:**
In our revised manuscript we will clarify that after removal of outliers and, arguably, calculating the maximum effects of fossil-fuel derived $CO_2$, we do not expect any other major biases in the $CO_2$ gradient (see also Figure in previous response). This implies, regardless of the uncertainty on the actual $CO_2$, the slope of the relation would not significantly change. However, the reviewer is correct in pointing out that this may influence our interpretation at what value of $CO_2$ $\epsilon p$ might become (in)sensitive to $CO_2$ changes. We will add further nuance to these statements in our revised manuscript and explain potential pitfalls in our approach. We also make it clearer that the 240 µatm – level is potentially a low estimate of (in)sensitivity and that, for practical reasons, the quadratic calibration should not be used below this level.

In addition, we will clarify that our initial Figure 5D and 5E we propagated (via Monte-Carlo analyses) a 5% error on the measured $CO_2$ and nutrients, as well as the standard error on the mean dinocyst $\delta^{13}C$. We erroneously omitted these uncertainties in Figure 5A-C and these will be added to the revised Figure. We will clarify our approach in the revised text.

Whilst the data presented here are interesting and important, the analysis at present is not sufficient to support the conclusions drawn robustly.

**Author response:**
We hope with the clarifications above, the calculated (maximum) uncertainty and the proposed adaptations of our revised manuscript we have alleviated the reviewers' concerns.

Technical corrections
22 use of "significantly" if this is meant in the statistical sense, please add p and n values, else reword.
**Author response:**
Changed to 'appreciable' as these data are not directly compared here.

24 ibid "significant"
**Author response:**
We will retain this statement but given the number of comparisons (20) we cannot provide p or n values of each comparison in the abstract of the manuscript. For significance we refer the reader to Figure 3 and keep the original generalised statement in the abstract.

40-42 This is a slightly eccentric choice of papers to cite here. At a minimum add an "e.g." but better to make it clear why these papers or a more comprehensive survey of the pCO2 proxy literature.
**Author response:**
We will include "e.g." before the cited references.

43 "However, many of the organic compounds used for CO2 reconstructions are not related to a single species, genus or even group." A fairly sweeping statement here not supported by any references. Which records and compounds are you referring to?

**Author response:**
We clarify in our revised manuscript this refers to proxy substrates in general (bulk organic matter and biomarkers such as phytane and alkenones) mentioned in the previous paragraph and we will refer to literature examples to illustrate this.

54 "extremely long-ranging" in space or time? Please be specific and it time list age range.
**Author response:**
We will add a statement to clarify this refers to *Operculodinium centrocarpum* and *Spiniferites* species as used here (see lines 65-67 for their age-range).

70 Should "cyst species" by cyst-forming species?
**Author response:**
To clarify we will rephrase this to 'εp derived from motile cells from controlled growth experiments can be translated to that of cysts formed in the natural environment.'

90 "Using standard palynological techniques" Please provide a citation.

**Author response:**
We will refer to Brinkhuis et al. (2003) who described the standard cold HCl/HF acid-digestion procedures employed to obtain palynological residues.

94 "ultraclean water" what is this? Ie quote a specific measure such as resistivity if reverse-osmosis teqnique has been used
95 "milliQ" is a brand name not a type of water. Please revise.
**Author response:**
We will change both these to demineralised water.

L104-5 is the 0.3-0.4 permil number precision or accuracy? How has accuracy been determined.
**Author response:**
We thank the reviewer for pointing out this unclear reference. We will remove 'accuracy' here as this statement was meant to refer to precision.

L343-4 "Badger, 2003, 2021;" These are two difference Badgers. Check BG style but likely need to include initials (lots, because they share first first name initial too).
**Author response:**
We thank the reviewer for highlighting this – we will revise to "Badger, M.R, 2003, Badger, M.P.S., 2021".

L 385. I'm not sure this is sufficient to meet the journal data policy. Pangaea doi should be available at publication.
**Author response:**
We will include the DOI as soon as that is available.

---

## Author Response (AR1)

**Response to reviewer 1**

General Comments
Frieling et al., present records of carbon isotope fractionation from the resting cysts of dinoflagellates to investigate their utility in reconstructing ancient atmospheric CO2. This record of core-top material advances earlier work based on laboratory cultures (and based on sound theoretical basis) and so brings the community closer to confidence that this proxy may work in environmental settings. They show there are differences in carbon isotope fractionation between different species, emphasising the importance of single-species records, and show greater 13-C depletion in their core-top samples compared to cultured, motile organisms. The paper is interesting and makes an important contribution, but some of the analysis is unsatisfactory due to uncertainty about the age of the individual cysts in the "core-top" samples (detailed below). Therefore without a thorough treatment of that uncertainty (which is currently lacking) it's difficult to know whether this proxy has utility. There are certainly hints that it does, but unfortunately this paper does not yet demonstrate that compellingly.

**Author response:**
We thank the reviewer for recognizing the potential importance of our work and the constructive criticism. In the response below and in our revised manuscript we have further clarified (1) how the carbon isotope data from individual cysts has been treated and (2) we further elaborate on the uncertainty in the age dating of the core-tops.

Specific Comments
The problem with using core-top samples is the substantial increase in atmospheric CO2 since the industrial revolution. As the authors note, it is highly uncertain whether the cysts are from the last week, the last year, decades or even centuries ago. The uncertainty around the contemporaneous CO2 is potentially very large. The "rough correction" to 1850 isn't really a correction at all, but an assumption which is not well supported, at best highly uncertain, and not really dealt with satisfactorily in the later analysis. The best approach (although expensive) would be to 14-C date some of these samples to see when this material actually dates to. The cheaper, and for this present study, more plausible approach would be to propagate through what is a really quite large uncertainty and see whether the conclusions still hold. Lines 147-8 state that "With the exception of pCO2, we hence assume all parameters (SST, SSS, nutrients) to be constant over the period the core top samples represent." A fundamental problem here is that the authors have little information (or at least present little data) about how long a period of time the core top samples do in fact represent. I'm not sure that the approach taken to this, systematically removing the most 13-C depleted samples is appropriate. Whilst it is certainly plausible that these individuals represent modern samples, the evidence is fairly circumstantial, and they could represent another confounding variable. What is the impact on the analysis if these samples are not removed?

**Author response:**
The reviewer comments on the potential of age-mixing of individual cysts in core-tops. This is very much a valid concern as we acknowledged in our original manuscript (lines 196-201). The main challenge here is that core-tops (the top-most 2 cm of sediment) contain individual sedimentary components with a range of ages. While it might be possible to $^{14}$C-date carbonate, bulk organic matter or even specific chemical components, these materials will derive from different times in the past not necessarily the same as the dinocysts analyzed here. Hence, a single measurement will not show the range of ages of the individual dinocysts

(i.e. the age-distribution of our individual dinocysts). Ideally one would date single-cell dinocysts using [14]C analyses but that is technically not feasible (yet).

As the reviewer correctly points out, alternatively one needs to show potential impact of the data-treatment, which we now include in a supplementary file to the revised manuscript (Supplementary Figure 1). In general, the impact of removing analyses based on exceptionally low amounts of C and exclusion of outliers has no appreciable impact on the regression parameters. The largest uncertainty indeed comes from the recent addition of anthropogenic carbon, the Suess effect. Comparing our calibration including and excluding the Suess effect is to our opinion therefore the best approach to estimate the maximum uncertainty in the regressions (lines 211-239).

[Figure]

**Supplementary Figure 1. Effects of data treatment on the difference between measured and adjusted $pCO_2$ and $\delta^{13}C$ ($\epsilon p$-$CO_2$).** Open symbols indicate measured $\delta^{13}C$, closed symbols represent data after eliminating small signals (<0.2 Vs) and outliers. Blue dots represent measured $CO_2$ values and red dots indicate the $CO_2$ around 1850 CE. For each dataset a simple linear regression, weighted to the number of measurements, is given. Dashed lines utilize measured $\delta^{13}C$, solid lines are from final $\delta^{13}C$ data. The red solid line is used in Figure 5A.

Supplementary Figure 1 illustrates the maximum error that may result from the Suess-effect by constructing a calibration with uncorrected data for each of the parameters (measured $\delta^{13}C$ and $CO_2$) and we compare this to the original calibration (solid red line). The main difference is in the intercept of the calibration, which is offset by the average difference in atmospheric $pCO_2$ between 1850 and the measuring date (~2000 CE; *ca.* 100 μatm) resulting from adjustment of measured $CO_2$ levels to pre-industrial times. The slopes of regressions using the data after drift-correction and outlier-removal are slightly shallower than those from the measured $\delta^{13}C$, but slopes of all regressions are statistically indistinguishable (lines 300-303) and the difference between the intercepts of the calibrations represents a worst-case scenario.

We also fully agree with the reviewer that it is critical to show that the data-treatment is free of potential preconceived biases regarding which values to include or exclude. We therefore further clarify in our revised manuscript how we first corrected for instrument drift at low C. This correction is exclusively based on repeat measurements of the IAEA-PE ($\delta^{13}C$ certified -32.15 ± 0.05‰) standard which show a convergence towards ~ -27‰ at the lowest intensities (original manuscript lines 172-174; see also Van Roij et al., 2017). After performing the drift correction, we subsequently identify outliers within the species-specific populations measured on a single sample (revised text line 226-231):

*Instead, we therefore illustrate the influence of $\delta^{13}C_{DINO}$ data treatment and $pCO_2$ correction (Supplementary Figure 1). For this, we compared both measured $pCO_2$ and $pCO_2$ around 1850 CE (see section 2.2) to $\varepsilon_p$ calculated using both our raw $\delta^{13}C_{DINO}$ data and the $\delta^{13}C_{DINO}$ data after drift-correction and removal of statistical outliers identified within the sample-specific single species populations.*

We find a greater proportion of negative outliers (original manuscript line 215) compared to positive outliers (hence a skewed data set; original manuscript Table 1). Since we correct using the PE standard and exclude both negative **and** positive outliers there is no assumption that the most negative are the most recent cysts (which one might assume based on the Suess effect but would be an *a priori* interpretation).

The reason why we think our data-treatment has some of a Suess-effect related bias which may exist between samples is as follows:
If we assume that (a) statistical distributions of $\delta^{13}C$ cyst populations are dominated by both $pCO_2$ and $\delta^{13}C-CO_2$ trends and that (b) these populations include a portion of both pre-industrial and recent times – the trends in both $pCO_2$ (higher $pCO_2$ > higher εp > lower $\delta^{13}Ccyst$) and $\delta^{13}C-CO_2$ (lower $\delta^{13}C-CO_2$ > lower $\delta^{13}Ccyst$) would impose a (negative) skew on the $\delta^{13}Ccyst$ populations. Prior to correction for skewing at low intensities and outlier removal some, but not all, samples show a negative skew, while after outlier-removal the vast majority is statistically indistinguishable from a normal distribution (Table 1 in the original manuscript). In addition to the supplementary figure showing the minor impact of excluding outliers (see above), we have now clarified this rationale in our revised manuscript (line 226-239).

Ie L 213-4 "We assume these assemblages are representative of ocean conditions prior to the massive increase in anthropogenic carbon emissions." This is very sizable assumption which whilst plausible is not currently supported by very much evidence.

**Author response:**
Please also see response above – we now provide a more elaborate reasoning why we assume that the vast majority of the analysed cyst populations may be considered to represent pre-industrial times or experienced only minor influence of anthropogenic $CO_2$. We will clarify in our revised manuscript that this is also based on the assumption that core-top material includes an age range, likely centuries up to millennia. This implies that in a core top sample only a very limited part of the population is derived from the most recent, anthropogenic times. For core-top samples a major impact of the Suess effect may be detectable only in the relatively few cysts that were formed in the past ~70 years (lines 211-225).

It appears from Figure 5a that no uncertainty at all has been applied to the assumed CO2 value – is this correct? This is not a fair assumption given the uncertain age of each cyst. In fact a plain reading of Figure 5a suggests that, rather than supporting a function between CO2

and ep apart from at <240 uatm CO2, ep is effectively constant, and only slightly higher above 310. Why therefore has 240 uatm been emphasised?

**Author response:**
The reviewer's comments prompted us to carefully revisit our analyses and numbers and led us to repair a few small errors and inconsistencies. As the data correction itself is a relatively minor adjustment (lines 236-239), any changes to the calibration equations and table 2 do not affect the final conclusions.

We clarify in our revised text that we do not expect any other major biases in the $CO_2$ gradient (see also Figure in previous response) after instrument drift correction and removal of outliers. The difference illustrated in the new supplementary figure 1 arguably shows a worst-case (maximum) effects of fossil-fuel derived $CO_2$ on the calibration. We therefore argue, regardless of uncertainty in the actual $CO_2$, the slope of the relation would not significantly change (lines 300-303). However, the reviewer is correct in pointing out that this may influence our interpretation at what value of $CO_2$ εp might become (in)sensitive to $CO_2$ changes. Accordingly, we added further nuance to statements on $CO_2$-insensitivity in our revised manuscript (lines 399-405) and better explain our approach regarding data treatment (paragraph 3.2.2 lines 206-250). We also make it clear now that the 240 µatm – level should be seen as the lower limit of (in)sensitivity and that, also for practical reasons, the quadratic calibration should not be used below this level (lines 299-300).

In addition, we clarify that for our original Figure 5D and 5E we propagated (via Monte-Carlo analyses) a 5% error on the measured $CO_2$ and nutrients, as well as the standard error on the mean dinocyst $\delta^{13}C$. We erroneously omitted error bars in Figure 5A-C and we did not provide a clear explanation of how errors were propagated for Fig. 5D and E. The revised figure now includes the error bars and the error propagation is properly explained in the Figure caption.

Whilst the data presented here are interesting and important, the analysis at present is not sufficient to support the conclusions drawn robustly.

**Author response:**
We hope with the clarifications above, the calculated (maximum) uncertainty and the proposed adaptations of our revised manuscript we have alleviated the reviewers' concerns.

Technical corrections
22 use of "significantly" if this is meant in the statistical sense, please add p and n values, else reword.
**Author response:**
Changed to 'appreciable' as these data are not directly statistically compared here.

24 ibid "significant"
**Author response:**
We have retained this statement - given the number of comparisons (20 for both variance and mean) we cannot provide p or n values of each comparison in the abstract of the manuscript. For significance of each comparison we refer the reader to Figure 3 and keep the original generalised statement in the abstract.

40-42 This is a slightly eccentric choice of papers to cite here. At a minimum add an "e.g." but better to make it clear why these papers or a more comprehensive survey of the pCO2 proxy literature.

**Author response:**
We have included "e.g." before the cited references.

43 "However, many of the organic compounds used for CO2 reconstructions are not related to a single species, genus or even group." A fairly sweeping statement here not supported by any references. Which records and compounds are you referring to?

**Author response:**
We have clarified this statement in our revised manuscript. The statement refers to proxy substrates such as phytane and alkenones mentioned in the previous paragraph (lines 43-45):

*"However, many of the organic compounds used for $CO_2$ reconstructions such as phytane (Witkowski et al., 2018) and alkenones (Pagani, 2013) are not related to a single species, genus or even group of organisms."*

54 "extremely long-ranging" in space or time? Please be specific and it time list age range.
**Author response:**
We have add the statement below (lines 54-57) to clarify this refers to the geological record of *Operculodinium centrocarpum* and *Spiniferites* species as used here (see lines 68-70 for their age-range).

*"The organic resting cysts from autotrophic species have excellent preservation potential, are often highly oxidation-resistant (Zonneveld et al., 1997, 2019; Kodrans-Nsiah et al., 2008) and several ubiquitous extant genera and species, such as Spiniferites spp. and Operculodinium centrocarpum, have extremely long geological records (Fensome et al., 1996; Williams et al., 2004)."*

70 Should "cyst species" by cyst-forming species?
**Author response:**
To clarify we have rephrased this to (lines 72-74): *'Although $\delta^{13}C_{DIC}$ exerts a major control on dinocyst $\delta^{13}C$ (Sluijs et al., 2018), it remains uncertain whether the $CO_2$ control on $\varepsilon_p$ of motile cells from controlled growth experiments can be translated to their cysts formed in the natural environment.'*

90 "Using standard palynological techniques" Please provide a citation.
**Author response:**
We now refer to Brinkhuis et al. (2003) who described the standard cold HCl/HF acid-digestion procedures employed to obtain palynological residues (line 93).

94 "ultraclean water" what is this? Ie quote a specific measure such as resistivity if reverse-osmosis teqnique has been used
95 "milliQ" is a brand name not a type of water. Please revise.
**Author response:**
We have changed both these to demineralised water.

L104-5 is the 0.3-0.4 permil number precision or accuracy? How has accuracy been determined.

**Author response:**
We thank the reviewer for pointing out this unclear statement. We have removed 'accuracy' here as this statement was meant to refer to precision.

L343-4 "Badger, 2003, 2021;" These are two difference Badgers. Check BG style but likely need to include initials (lots, because they share first first name initial too).
**Author response:**
We thank the reviewer for highlighting this – we have revised to "Badger, M.R, 2003, Badger, M.P.S., 2021".

L 385. I'm not sure this is sufficient to meet the journal data policy. Pangaea doi should be available at publication.
**Author response:**
We have included the Mendeley data DOI and will release the embargo upon publication.

**Response to reviewer 2**

The work by Frieling and colleagues is strong framework and a much-needed study that will open a new opportunity for applications of organic microfossil 13C analysis. Like single species foram analyses (the benchmark for modern carbon and oxygen isotope studies) single or several organic microfossil 13C limits the breadth of sources to sedimentary organic matter and limits the degrees of freedom in a highly advantageous way. This study is the gateway to the deeper geologic record that will allow broad application of the dinocyst proxy to ancient carbon cycle studies. The questions below are meant to enhance the discussion, but the work, as it is, stands on its own as it is presented.

**Author response:**

We thank the reviewer for the positive and constructive review of our work. The review highlights a number of aspects of the methodology which often do not appear in the published literature but in this case will be helpful to provide a baseline for further work.

From a methodological perspective I appreciate the details provided here. Controlling for size and process length is great approach but do you see relationships between $d^{13}C_{cyst}$ and cyst size?

**Author response:**

The reason we aim to exclude size-dependent $\delta^{13}C$ differences is that for e.g. foraminifera, coccoliths and living dinoflagellates a size-dependent $^{13}C$ fractionation has been observed previously (Burkhardt et al., 1999; Hoins et al., 2015). Unfortunately, a size-dependent $\delta^{13}C$ relation in modern ocean dinocysts is beyond what we can reasonably test with our method – because of the analytical uncertainty when measuring such small (30-40 µm diameter) individuals the number of required repeat measurements would become unpractically large. However, we fully agree this is a logical next step, once the methodology is sufficiently developed to achieve the precision needed to distinguish between individual cysts' $\delta^{13}C$ signature of modern species.

To what degree do you feel that the time averaging affected your data? Do you have access to any 14C dates of the surface sediments? From here you could potentially model the expected range of 13C values of DIC accounting for Suess Effect. More details in the manuscript on your rough correction would be helpful.

**Author response:**

This point is in line with one of the points raised by the other reviewer. We elaborate on our reasoning regarding age-control below and in the revised manuscript (lines 139-148).

We fully agree that better age-control on sediments, but especially of the individual cysts would be helpful and also modelling $\delta^{13}C_{DIC}$ (subtracting the Suess Effect) as the reviewer points out is a step we would like to take if feasible. However, the use of $^{14}C$ dates (often carbonate, otherwise bulk or macro-scale organic matter) is complicated as they cannot be measured on the cysts themselves – not even when concentrating large amounts of cysts. Therefore, these analyses cannot represent the true dinocyst age, and certainly not an age distribution as the individual cysts in our data represent. We feel that incorporating any

sample 'age' correction for now would mainly result in erroneous corrections and hence we prefer to illustrate the range in $CO_2$ and $\delta^{13}C$ corrections (a maximum error range), which we elaborate on in our revised manuscript (paragraph 3.2.2, lines 211-239).

Specifically, the maximum influence of the Suess-effect is assessed by constructing calibrations with uncorrected data for each of the parameters (measured $\delta^{13}C$ and $CO_2$). These are compared these to the original calibration (see Figure below, which we added as Supplementary Figure 1). We stress the difference between these calibrations is a worst-case scenario (i.e. maximum offset).

[Figure]

*The above figure has been added to the supplementary information. The figure shows the offset between measured and adjusted $pCO_2$ and $\delta^{13}C$ ($\varepsilon p$-$CO_2$) values. Open symbols indicate measured $\delta^{13}C$, closed symbols represent data after eliminating small signals ($<0.2$ Vs) and outliers. Blue dots represent measured $CO_2$ values and red dots indicate the $CO_2$ around 1850 CE. For each dataset a simple linear regression, weighted to the number of measurements, is plotted. Dashed lines are based on measured $\delta^{13}C$, solid lines are from final $\delta^{13}C$ data. The regression plotted as a red solid line is used for Figure 5A.*

What do you think is the background blank source? Is it from atmospheric aerosols that adhere to all surfaces regardless of precautions or is it from within the nickel plate? (Does the nickel plate show scoring from the laser?). Regardless, the approach to signal size to noise, considerations of the blank and other corrections seem reasonable. These consideration are important not only for your study and approach but for the future potential of this kind of analysis for sample return from Mars and elsewhere.

**Author response:**

We have expanded our methods section with a few statements covering these aspects (line 175-189):

The origin of the blank source is currently unknown and proved difficult to constrain. When setting up the system, we used a liquid $N_2$ cooled trap to pre-concentrate $CO_2$, before releasing it to the IRMS. With that system inherently the 'blank' $\delta^{13}C$ was also much larger, as it also concentrates the blank signal, which is why we returned to the current true continuous flow system. However, even with that concentrated blank it remained difficult to constrain its C-isotopic value and therefore also the opportunity to confidently identify the source (see also e.g., Van Roij et al., 2016). However, the trapping experiment provides some useful constraints on the potential size of the background contamination, which has been added to the revised manuscript (lines 185-189).

We considered several potential sources: (1) atmospheric $CO_2$ (air or particles that come into the system when opening the sample cell), (2) residual material from earlier measurements (wall sorption) and (3) possibly a minor amount of additional C from the water the cysts are isolated from. We assume that the first source are atmospheric particles the reviewer hints at. The nickel plate is not ablated with the low energy densities used and testing with clean plates (before any sample is added) shows the nickel plate itself does not add to the C signal, Similarly, repeated test with water droplets being added showed this to be a minor carbon source (< 10% of total blank). Etching or sample water also would not influence the PE standard as the ablation does not fully penetrate the plastic and no water is used for preparing the PE standard. In addition some micro leakage of the system at connections for the GC and/or the ovens could also add to the blank signal. Because it is currently impossible to constrain the source we here prefer to refrain from speculation on potential blank sources, other than stated above.

Regardless of the source, the combined contribution of these factors proved to be minor and stable, and hence we were able to correct for it (as can be seen in Fig. 2).

Line 280: From this discussion I think I favor your argument that intercyst variability reflects individual differences. One can envision that individual cells or cysts have significantly different 13C values owing to the randomness of cellular growth, changes in microenvironments of growth that also affect DIC and CO2 13C. Add in the time averaging from core top sample collection it is not a surprise that you see large variance. In fact, I would be worried if you did not. Your suggestion of controlling for size, as much as one is able, is a good idea.

**Author response:**

We thank the reviewer for their view – indeed this is also our preferred scenario for explaining the intra-sample variability. We have included the reviewer's points on potential for $^{13}C$-impact of cell-microenvironments and growth-induced randomness to the $\delta^{13}C$ of the cyst in our revised manuscript (lines 318-319).

Line 280: For standards have you considered dissolving a standard material like caffeine in water and allowing it to dry onto a surface and analyzing that (you could spray it or something). At the very least here you could assume that the starting composition is isotopically uniform. I supposed 13C differences could arise from the drying process, but it may be better than PEF.

**Author response:**

We have been in search for a sufficiently homogenous standard with similar ablation and material characteristics as the polyethylene plastic currently used. We prefer a more or less similar material as potential differences in ablation characteristics could interfere with our method of standard bracketing in which we compare the signal of the ablated dinocysts with that of the standard. A standard material with similar ablation characteristics is also preferable from an operational perspective. For example, if the sample plate is covered entirely with a standard material (as indeed could be done with spray or by submerging), it becomes difficult to insert the sample without contaminating the system. Moreover, we prefer a standard with a relatively low vapor pressure as a somewhat volatile standard material in the sample holder could increase the blank signal. However, we also acknowledge that solid standard material (e.g., a film or foil) is inherently hampered by inhomogeneities on the scale required (~80 μm spot size). We are involved in the constant quest for better and improved standards (see e.g Boer et al., 2022* in which we developed a new standard using micro-milled powders), but this is challenging for solid organics. When cooling or drying (crystallisation) organic substances (e.g., glycerine, various corn-starch based products, monosodium glutamate) we observed the solid to structurally differentiate and no homogeneity could be reached. Work on a new standard continues (including the suggestion given by the reviewer using spraying) and, once successful, we will implement such a standard in our methods and report on it.

*Boer, W.; Nordstad, S.; Weber, M.; Mertz-Kraus, R.; Hönisch, B.; Bijma, J.; Raitzsch, M.; Wilhelms-Dick, D.; Foster, G.L.; Goring-Harford, H.; Nürnberg, D.; Hauff, F.; Kuhnert, H.; Lugli, F.; Spero, H.; Rosner, M.; van Gaever, P.; de Nooijer, L.J.; Reichart, G.-J. (2022). New calcium carbonate nano-particulate pressed powder pellet (NFHS-2-NP) for LA-ICP-OES, LA-(MC)-ICP-MS and μXRF. *Geostand. Geoanal. Res. 46(3)*: 411-432.

Line 300: Have you investigated the compositional differences between cyst and motile cells? I am familiar with the references you report on this issue but what specifically are the differences? What proportion of the carbon from the cell transferred to the cyst? Is this known?

**Author response:**

These are important outstanding questions and subject of currently running as well as planned work regarding cell compartment derivation of cyst molecules using LC-IRMS, cyst production – excystment experiments to assess cell to cyst fractionation. In short, for this we need to compare the core-top cysts to cultured motile cells to ensure that cells and cysts can be related one on one, which is a line of research in itself.

---

## Author Response (AR2)

Dear Joost and co-authors,

First, my apologies for the long delay in reaching a decision. Your revised version was re-evaluated by one of the original reviewers, and while your revisions have addressed several earlier comments, they still find that insufficient support for the current conclusions is present; mainly due to the uncertainties in the dates of your core-top samples.
This is a valid concern which indeed remains an important caveat of the manuscript- although I understand there is no easy way to resolve this and provide dates with confidence.
-Consider if it would be possible to constrain sedimentation rates for the sites more quantitatively, and in particular then also whether large differences in sedimentation rates are present.
-as noted also by this reviewer, the supplementary figures show that many of the relationships are driven by 3 data points (relationships with pCO2 and/or nutrients); yet it is not clear to the reader which sites these data correspond to.

I feel your data are valuable and use cutting-edge methodologies and therefore merit giving an additional opportunity for revisions; but leave it to the authors to decide whether they feel they can go further in accommodating the reservations of Reviewer #2. In case not, you could consider reformulating your discussion and conclusions in a more cautious way and add some discussion on the way forward to explore the controls on dinoflagellate cyst d13C in future studies.

With best regards
Steven Bouillon

**Author response:**
We once again thank the editor and reviewer for their constructive views on our work.

Indeed, we agree it is unfortunate that we have no means of constraining sedimentation rates for these sites. But as mentioned, even if we did have sedimentation rates, that would still be an 'imperfect' approximation of cyst-production age as the age-distribution of the dinocysts themselves cannot be constrained. This is a challenge unique to the here presented data and we fully agree that these challenges should be discussed in the text and that, in absence of constraints, some additional sensitivity analyses are needed and these will be added to the revised manuscript.

As suggested by reviewer 2, we explore the potential impact of an offset between pre-industrial $p\text{CO}_2$ and potentially 'modern-like' $p\text{CO}_2$ cysts. As suggested, we propagate a 45±15 ppm error (a normal distribution with a 3-sigma range between ~0 and ~90 ppm) and the analytical errors (5% of the $p\text{CO}_2$ value). In a second scenario we explore the errors associated with a random draw $p\text{CO}_2$ -change from pre-industrial, assuming cysts were produced between 1800 – 2000 CE. These $p\text{CO}_2$ values are strongly non-normally distributed, and might be considered a more realistic scenario, as the exponential $p\text{CO}_2$ rise over the last decades implies there is a greater likelihood of cysts being produced during times of only moderately elevated $p\text{CO}_2$ conditions. Both scenarios result in an offset of the absolute $p\text{CO}_2$ values, but regression parameters are fairly robust with regard to these errors. Still, the offset and slight changes in the regression parameters do imply that the $p\text{CO}_2$ estimates resulting from the original 1850 CE assumption are more likely to underestimate both absolute values and $p\text{CO}_2$ variability, which will be added to the revised manuscript.

Furthermore, we include a secondary dataset spanning 0 – 1500 CE from the North Atlantic offshore Ireland (Feni Drift; Richter et al. 2009). The measured $\delta^{13}C_{DINO}$ in these sediments are broadly similar to those of the nearest three core-top samples. When the core-tops are grouped together, they fall exactly on the drift-sediment average, as do the data distribution and $^{13}C$-variance within the cyst population. This shows that an anthropogenic imprint on those three core-top samples cannot be detected. Clearly, however, the potential for added uncertainty for other localities can still not be fully dismissed.

The above challenges and statistical exploration are now discussed in a separate section "*4.5 Challenges of age-control and potential caveats associated with anthropogenic carbon*". In addition, Figure 6 now includes a plot that should ease identification of the localities used in this study.

Unfortunately, my primary concern with this paper remains, and the additions and alterations do not go far enough for me to be confident that the data support the conclusions. The addition of Supp Figure 1 (which should be in the main paper) does demonstrate the relatively minor impact of the removal of "outliers" however it also highlights the more substantial concern.

Firstly, on the removal of "outliers" the selection of these is based on an a priori assumption that CO2 does control ep, and that there are some modern contaminating cysts in the samples. However the a priori nature of this correction means it is arguably inappropriate, and the selection process fairly arbitrary.

**Author response to point 1:**
We are happy to see reviewer agrees the impact of outlier omission is very minor. However, we are surprised that the reviewer believes $CO_2$-dependency is an a priori assumption to outlier detection or omission; this is not the case. We further clarify that outliers are identified from the $\delta^{13}C_{DINO}$ data populations, for each sample and species individually, as indicated in the manuscript and also described in our reply to the previous review of Reviewer #2 (lines 238-240):

*"This final step of data-treatment removed positive and negative measurement outliers from the sample- and species-specific $\delta^{13}C$ population (outside ±2.5 IQR), after eliminating the extremely low-signal intensities (<0.2 Vs) and correcting for the drift induced by background C in the system."*

While negative (*i.e.* $\delta^{13}C_{DINO}$ below the sample- and species-specific population average) outliers (*n* = 21) clearly outnumber positive outliers (*n* = 3), both are identified. After outlier omission a greater number of distributions are indistinguishable from a normal distribution; which is a common effect of omitting outlier values. In fact, the data correction, exclusion of measurements with low signal intensity and outliers leads to somewhat poorer, not better, correlation to $pCO_2$. We have carefully revisited our explanation of the selection process so that it is now clearer that this is not arbitrary nor biased (line 221-227):

*"Based on typical deep ocean sedimentation rates in the range of centimetres per kyr, the core-top samples are expected to contain a mixed assemblage of dinocysts produced mostly within the last centuries to millennia but could also include cysts produced during the last few decades that are likely affected by anthropogenic influences. It is particularly relevant to consider because a steep $\delta^{13}C$ decrease (~2‰ since 1850 CE of which >1.5‰ occurs after 1950 CE) (Francey et al., 1999; Keeling et al., 2017) accompanies the $pCO_2$ rise (>130 ppmv since 1850 CE, of which >100 ppmv after 1950 CE). So even if enhanced carbon isotope fractionation at higher $pCO_2$ (Freeman and Hayes, 1992; Hoins et al., 2015; Brandenburg et al., 2022) would not play a role, the most recent specimens are likely to be impacted by decreasing $\delta^{13}C_{DIC}$."*

And lines 249-254:

*"Distinctly non-normally distributed $\delta^{13}C$ values were not previously observed in recent pollen and ancient dinocyst species analyzed with the same method (van Roij et al., 2016; Sluijs et al., 2018). The here presented down-core pre-industrial $\delta^{13}C_{DINO}$ show a similar mean, variance and data distribution to the nearby core-top samples (Supplementary Fig. 1), suggesting that, at least for these nearby localities, the analysed core-top specimens represent pre-industrial conditions. We find an influence of Suess-effect and increased $pCO_2$ impacts on*

*the $\delta^{13}C_{DINO}$ data is the most likely factor to explain the appearance of a small number of predominantly $^{13}C$-depleted outliers and resulting (subtle) negative skewing of the $\delta^{13}C$ distributions (Fig. 4)."*

Secondly the acknowledgement that these core tops are of mixed age both in the need to remove "outliers" and the statement at the start of the response: "core-tops (the top-most 2 cm of sediment) contain individual sedimentary components with a range of ages. While it might be possible to 14C-date carbonate, bulk organic matter or even specific chemical components, these materials will derive from different times in the past not necessarily the same as the dinocysts analyzed here. Hence, a single measurement will not show the range of ages of the individual dinocysts (i.e. the age-distribution of our individual dinocysts)." highlights that the age of the material analysed is highly uncertain.

> **Author comment:** Please note that we already acknowledge the age uncertainty in previous versions of the manuscript and have expanded this discussion based on the valuable comments of reviewer #2 in the previous and the current version of the manuscript.

In many cases in oceanography this is the case but not consequential, as the potential difference mixed ages causes in an individual core top sample is much less than the signal that is being reconstructed. This is not the case here. Whilst it is not unusual (although strictly wrong) to assume that a core top is "present day" in this paper it is accepted that the core top is a time-integrated slice, but then assumed that it all represents an arbitrary date of 1850.

> **Author response:** This is not our assumption. In our analyses, we use the date of (pre-)1850 simply to distinguish cysts affected by fossil fuel combustion and those older than that for our sensitivity studies.

As Supp Figure 1 and Figure 1 show, the signal being reconstructed – the different between lowest and highest CO2 at the different sites, is on order 150 ppm, whilst the difference between modern and 1850/pre-industrial CO2 is close to 90 ppm. As these are similar order of magnitude, the tightness of the age control becomes a major concern as any contamination of modern specimens in "low CO2" high latitude sites could overwhelm the signal.

At present it is impossible to assess how great a problem this could be. Estimates of site sedimentation rates are not provided (beyond "typical deep ocean sedimentation rates in the range of centimetres per kyr" lines 314-5) and the sites are not sufficiently documented or cited in a way that allowed me to find out whether sedimentation rates for the sites are available, and whether they are particularly low, high, or variable across the calibration set. At present, the 2 cm core-tops may represent 200 years of time, during which the variable they are trying to reconstruct has changed substantially. This uncertainty is not sufficiently dealt with in the calibration, with a 5 % uncertainty added but no explanation as to how this value was reached.

> **Author response:**
> The 5% error derives from the analytical error on the $CO_2$ measurements, and this has now also been clarified in the main text (line 148-149):

*"We employed a Monte Carlo simulation to assess the potential impact of the $pCO_2$ correction by propagating (1) the 5% analytical error on $pCO_2$ values [...]"*

A more robust treatment of these data would be to assume a 90 ppm (present to 1850 adjustment) uncertainty in the pCO2 concurrent with cyst formation and rerun the regression analyses with this uncertainty included. Better still, for each site an assessment of the age uncertainty for the core top could be made (based on sed rate) and a CO2 uncertainty for the population of individuals estimated. This should then be made used in the regression analysis.

Without this step, it is impossible to know whether the regressions proposed are plausible.

**Author response to point 2:**
We thank the reviewer for challenging us to think more critically about this step in our analyses. The sedimentation rates for the core material are unfortunately unconstrained but, using explorative error analyses and new data (see below), we now show unlikely of importance to the main conclusions ($pCO_2$ and nutrients control $\varepsilon_p$ in dinocysts).

To accommodate the suggestion of the reviewer, our revised manuscript includes two new scenarios assuming the top sediment represents 200 years where we explore 'worst-case' conditions and their potential effects on the calibration. The first scenario adds a $pCO_2$ value from a single random draw on a sample-level, from a normal distribution that spans $0 – 90$ ppm to the analytical error (5%). The second scenario is identical but uses a random resampling of $pCO_2$ values above pre-industrial $pCO_2$ levels for the period $1800 – 2000$ CE. The assumption that the core-top records 200 years of sedimentation is equivalent to a 2 cm core-top slice at a very high-accumulation-rate site (10 cm/kyr sedimentation rate).
These analyses show that, while the reviewer is correct to state that the $CO_2$ difference between the date of measuring or collection and the hypothesized bulk of the data (1850 CE and older) is substantial, it is unlikely that "*any contamination of modern specimens in "low $CO_2$" high latitude sites could overwhelm the signal*". When adding a $0 – 90$ ppm error to all data or resampled $pCO_2$ values, with exception of the absolute values, the regression parameters are fairly robust.

In addition to the ~90 ppm error scenario, we also simulate the uncertainty in sedimentation rates by assigning a $pCO_2$ value (above pre-industrial) sampled from the atmospheric $CO_2$ between 1800 and 2000 CE. The reason for doing this is that the $pCO_2$ values from 1800 to 2000 follow a non-normal distribution and therefore strongly biased (anthropogenic) values are considerably less likely to occur. Errors are included in the regressions through a simple resampling of atmospheric $pCO_2$ for each datapoint. Both analyses and error distributions are included in the new Figure 7 (included below).

[Figure]

***Figure 7. Data treatment and potential effects of anthropogenic carbon emissions. A.** Effects of data treatment on the difference between measured and adjusted pCO₂ and δ¹³C (εp-CO₂) (same as Fig. 3). Open symbols indicate measured δ¹³C, closed symbols represent data after eliminating small signals (<0.2 Vs) and outliers. Blue dots represent measured CO₂ values and grey dots indicate the CO₂ around 1850 CE. **B.** Quadratic regression (red line in Figure 6B) with propagated analytical error on pCO₂ and δ¹³C only, using CO₂ values around 1850 CE (grey filled symbols in panel A. **C.** As in B but with addition of a 45 ± 15 ppm error to reflect potential impact of anthropogenic CO₂ in orange. Grey dots and curve of panel B are added as a comparison. **D.** As in B but with addition of the CO₂ increase relative to pre-industrial in the period 1800 – 2000 CE. Insets (bottom right) in panels B, C and D show the combined error distributions (in ppm) imposed on pCO₂. All error bars in panels B – D on pCO₂ and εₚ-CO₂ are 2.5 – 97.5% percentile ranges from Monte Carlo simulations (n=1000).*

In addition, in our revised manuscript we now include a new down-core $\delta^{13}C_{DINO}$ *(O. centrocarpum)* dataset spanning 0 – 1500 CE and compare this data to $\delta^{13}C_{DINO}$ of the same species in three nearby core-top samples. The $\delta^{13}C_{DINO}$ distributions of the pre-industrial cysts match those of the core-top samples and provides circumstantial evidence that the nearby core-tops (still) include mostly cysts unaffected by anthropogenic sources (line 250-252). Histograms of these data are included below and as a new Supplementary Fig. 1.

[Figure]

*Supplementary Figure 1. Histograms of down-core $\delta^{13}C_{DINO}$ (O. centrocarpum) for ENAM9606 in the North Atlantic compared to three nearby core-top samples (PE360-24, PE360-45, ENAM9609b). ENAM9606 (55.650 ºN, -13.985 ºE) represents down-core $\delta^{13}C_{DINO}$ for ~0 – 1500 CE (Richter et al., 2009), whereas PE360-24 (55.496 ºN, -15.801 ºE), PE360-45 (55.539 ºN, -15.845 ºE) and ENAM9609b (57.160 ºN,-10.26ºE) represent nearby core-top $\delta^{13}C_{DINO}$. Frequency (y-axis) indicates the number of measurements for each of the 1‰-wide $\delta^{13}C$ bins. All $\delta^{13}C$ distributions are background-corrected values, without outliers.*

Finally, we agree with the reviewer that further details and error propagation would benefit further assessment of this work and utilize the above analyses to illustrate where the main uncertainties lie to inform future efforts. Accordingly, we have added a paragraph in the discussion "*4.5 Challenges of age-control and potential caveats associated with anthropogenic carbon*" (lines 429-448). This includes a brief outline of the challenges and the results of the statistical exercises. We hope to have sufficiently illustrated the potential for age-dependent errors to play a role in the calibration offered in our work and following the reviewer's suggestion, we have added further nuance where needed.

Overall, we find the various 'error' scenarios indicate that the calibration based on pre-industrial $pCO_2$ is fairly robust, though the absolute values and perhaps variability may be underestimated, if a significant number of cysts was produced (long) after 1850 CE.

Our explorative error analyses yielded the following changes to equation 2a, which have been included in our revised manuscript (lines 291-302):

Equation 2 quadratic (only suitable for use > 240 µatm)
$$\varepsilon_{p-CO2} = 40.8 \pm 7.2 - 0.23 \pm 0.055\ pCO_2 + 4.88 \pm 1 \times 10^{-4}\ pCO_2{}^2$$
(Adjusted $R^2 = 0.79$, p <0.001, RSME = 1.13 ‰) (Figure 6B,F)

Equation 2b quadratic (Monte Carlo constrained errors – analytical for $pCO_2$ and $\varepsilon_{p-CO2}$) (Figure 7B)
$$\varepsilon_{p-CO2} = 35.6\ ^{+5.8}/_{-5.6} - 0.19\ ^{+0.045}/_{-0.045}\ pCO_2 + 4.1\ ^{+0.91}/_{-0.88}\ 10^{-4}\ pCO_2{}^2$$

Equation 2c quadratic (as 2b with additional $45 \pm 15$ ppm $pCO_2$ error) (Figure 7C)
$$\varepsilon_{p-CO2} = 39.3\ ^{+11.5}/_{-8.8} - 0.19^{+0.058}/_{-0.076}\ pCO_2 + 3.4\ ^{+1.3}/_{-0.95} \times 10^{-4}\ pCO_2{}^2$$

Equation 2d quadratic (as 2b with resampled $pCO_2$ rise 1800 – 2000 CE) (Figure 7D)
$$\varepsilon_{p-CO2} = 29.8\ ^{+11.0}/_{-8.0} - 0.13\ ^{+0.061}/_{-0.084}\ pCO_2 + 2.6\ ^{+1.5}/_{-1.1} \times 10^{-4}\ pCO_2{}^2$$

Minor comments.

L 43-45 "However, many of the organic compounds used for CO2 reconstructions such as alkenones (e.g. Pagani, 2013), phytane (e.g.Witkowski et al., 2018), porphyrins (e.g. Freeman and Hayes, 1992) or bulk organic matter (e.g. Hayes et al., 1999) are not related to a single species, genus or even group of organisms." It is incorrect to say that alkenones are not related to a single group of organisms.

**Author response:**
Thanks for pointing this out. We have rephrased to "*are not unique to a single species, genus and sometimes not even a group of organisms.*" (lines 44-45)

L 132 Is there a confusion between Ocean Data Viewer and Ocean Data View here?

**Author response:**
Yes, this should read Ocean Data View and not '*viewer*' (link also corrected) (line 134).

L 140 Assumption of moderate to low sedimentation rate of < 10 cm kyr given. Is this at all reasonable for the sites used? Are any sites particularly low or high sedimentation rate? Note that at this sed rate the 2 cm core tops span 200 years of accumulation.

**Author response:**
As the sedimentation rates are unconstrained, we unfortunately must rely on generalisations. As all our sites are reasonably far offshore and not deposited in drift sediments, we find an upper limit of 10 cm/kyr is reasonable (line 141-143).

L 145 "this correction has only a small impact on the patterns in the CO2 data (Supplementary Figure S1)." This is not a fair assessment of what Figure S1 shows. It shows that the difference in regression between adjusting and not adjusting for "eliminating small signals and outliers" is small, but the impact of assuming all 1850 CO2 vales is very large.

**Author response:**
We have clarified that the offset is potentially large (line 145-147) but the regression parameters are fairly robust (lines 320-325).

What do the difference sizes of symbol on Supp. Fig 1 mean? Can error bars please be added.

**Author response:**
The symbol size represents the number of measurements included from each location, similar to those in Figure 6 – this has been added to the figure key (note that SI Fig. 1 is now included as Figure 3 and 7A). Error bars can be added but will make this plot very crowded; in addition to the 76 data points with different sizes and fill, it will then include x- and y-error bars of both different color and type (see Figure below, left). We understand the value of included error bars but find the version without (Figure below, right) is a clearer illustration of how the data treatment affects the data used in the regressions. Of course, we are willing to optimize a version with error bars if that is deemed preferable.

[Figure]

Supplementary Figure 1 and Figure 5 also highlight that most of the regression is driven by three samples with high estimated CO2. As a full dataset is not provided it's difficult to know for sure, but from Figure 1 I estimate these are all from the Mediterranean. Is there a difference between Mediterranean and Atlantic types? Is there any reason to think that the correction may be more or less valid in the Mediterranean? What are the sedimentation rates for these and all sites? Not enough information is provided in the manuscript or provided files for me to chase down the individual core documentation.

**Author response:**
We thank the reviewer for this suggestion - to facilitate identification of individual samples in the fractionation plots, site numbers are now included in Figure 6A, which correspond to the now numbered sites listed in Table 1 and Figure 1.

The regression is most dependent on the cluster of data points as well as the three data points referred to by the reviewer. The two highest fractionation values derive from the Mediterranean while the third highest fractionation value is observed in the core-top sample from the equatorial Atlantic. The other Mediterranean sample plots within the larger cluster of values.

No changes in cyst morphology were seen between samples and we thus have no reason to assume that the cysts from the three localities with the highest fractionation values or those from the Mediterranean sites differ from the majority of the North Atlantic localities.